# Therapeutic manipulation of *IKBKAP* mis-splicing with a small molecule to cure familial dysautonomia

Masahiko Ajiro[1,2], Tomonari Awaya [2], Young Jin Kim[3], Kei Iida [4], Masatsugu Denawa [4], Nobuo Tanaka[4], Ryo Kurosawa [2], Shingo Matsushima[1,2], Saiko Shibata[1,2], Tetsunori Sakamoto[2], Lorenz Studer [5], Adrian R. Krainer[3] & Masatoshi Hagiwara [1,2✉]

Approximately half of genetic disease-associated mutations cause aberrant splicing. However, a widely applicable therapeutic strategy to splicing diseases is yet to be developed. Here, we analyze the mechanism whereby *IKBKAP*-familial dysautonomia (FD) exon 20 inclusion is specifically promoted by a small molecule splice modulator, RECTAS, even though *IKBKAP*-FD exon 20 has a suboptimal 5′ splice site due to the IVS20 + 6 T > C mutation. Knockdown experiments reveal that exon 20 inclusion is suppressed in the absence of serine/arginine-rich splicing factor 6 (SRSF6) binding to an intronic splicing enhancer in intron 20. We show that RECTAS directly interacts with CDC-like kinases (CLKs) and enhances SRSF6 phosphorylation. Consistently, exon 20 splicing is bidirectionally manipulated by targeting cellular CLK activity with RECTAS versus CLK inhibitors. The therapeutic potential of RECTAS is validated in multiple FD disease models. Our study indicates that small synthetic molecules affecting phosphorylation state of SRSFs is available as a new therapeutic modality for mechanism-oriented precision medicine of splicing diseases.

[1] Department of Drug Discovery Medicine, Kyoto University Graduate School of Medicine, Kyoto, Japan. [2] Department of Anatomy and Developmental Biology, Kyoto University Graduate School of Medicine, Kyoto, Japan. [3] Cold Spring Harbor Laboratory, Cold Spring Harbor, NY, USA. [4] Medical Research Support Center, Kyoto University Graduate School of Medicine, Kyoto, Japan. [5] Center for Stem Cell Biology, Sloan Kettering Institute, New York, NY, USA. ✉email: hagiwara.masatoshi.8c@kyoto-u.ac.jp

Splicing mutations through the creation or disruption of splicing cis-elements induce constitutive gain or loss or the alternative splicing of target exons, which in turn prevent functional protein production by generating stop codons or frameshifts. Pre-mRNA splicing is catalyzed by five essential small nuclear RNAs (U1, U2, U4, U5, and U6 snRNA) and more than 100 proteins. In addition to these essential factors (e.g., U1 small nuclear ribonucleoprotein (snRNP) to recognize the 5′-splice site and U2 snRNP to recognize the branchpoint sequence), various trans-acting splicing factors are recruited to cis-regulatory elements to orchestrate alternative splicing. Several approaches have been taken to amend mutation-derived splicing defects by targeting either cis-regulatory elements or trans-acting splice factors. The former approach includes the antisense-oligonucleotide drug nusinersen, which rescues deficient SMN2 exon 7 inclusion for spinal muscular atrophy therapy[1–3], and small-molecule compounds, such as SMN-C3 and RG7916, that also promote SMN2 exon 7 inclusion, in part by directly binding to an imperfect RNA helix formed by the exon 7 5′-splice site and U1 snRNA acting for bulge repair[4–8]. Examples of the latter approach include small-molecule inhibitors for CDC-like kinase (CLK) or serine/arginine-rich protein kinase (SRPK)[9], which then inactivate serine/arginine-rich splicing factors (SRSFs) through serine phosphorylation in their carboxy-terminal RS domain. Previously, we synthesized CLK inhibitors[10] and demonstrated their therapeutic potential in disease models of cystic fibrosis (c.3849 + 10 kb C > T mutation of cystic fibrosis transmembrane conductance regulator (CFTR))[11], Duchenne muscular dystrophy (c.4303 G > T mutation of dystrophin)[12], and anhidrotic ectodermal dysplasia with immunodeficiency (EDA-ID; IVS6 + 866 C > T mutation of NF-kappa-B essential modulator (NEMO))[13].

In parallel, we developed a dual-color splicing reporter system combining two different fluorescent proteins to identify both cis-elements and trans-acting factors involved in targeting alternative splicing events in worms, mice, and cultured cells[14–16]. Recently, we applied this system to screen a chemical library and found a small-molecule RECTAS, which corrects abnormal splicing caused by the IVS20 + 6 T > C splicing mutation of the (inhibitor of κ light polypeptide gene enhancer in B-cells, kinase complex-associated protein (IKBKAP) gene[17]. The homozygous mutation is mainly found in the Ashkenazi Jewish population and is responsible for >99.5% of cases of familial dysautonomia (FD) (also known as Riley-Day syndrome or hereditary sensory/autonomic neuropathy type III)[18–20]. FD is an autosomal recessive genetic disease, and autonomic neurons and peripheral sensory neurons derived from neural crest cells are impaired in development, function, and survival[18–20]. Patients with FD exhibit various symptoms including cardiovascular instability, recurrent pneumonia, vomiting/dysautonomic crisis, gastrointestinal dysfunction, decreased sensitivity to pain and temperature, and defective lacrimation. The IVS20 + 6 T > C mutation causes abnormal IKBKAP exon 20 skipping, due to impaired recognition of the mutant 5′ splice site by U1 snRNP[21,22], leading to a frameshift and the generation of a premature termination codon in exon 21 of IKBKAP mRNA[20]. This frameshift in turn results in reduced expression of IKAP (or ELP1), compromising tRNA modification and neuronal cell survival[17,23,24]. RECTAS directly affects mRNA splicing, and our transcriptome analyses revealed that RECTAS affects the splicing of only a limited set of genes, suggesting that it has high specificity and a direct role in correcting IKBKAP aberrant splicing[17]. Sequence analysis of the responsive exons suggests that RECTAS promotes exon recognition by either inhibiting splicing repressors, such as heterogeneous nuclear ribonucleoproteins (hnRNPs), or stimulating splicing activators. RBM24 functions as a tissue-dependent splice activator by binding to a cryptic intronic splicing enhancer

element (IVS20 + 13 to +29) downstream of the intronic 5′ splice-site mutation of IKBKAP-FD and promotes U1 snRNP recognition of the mutant 5′ splice site[25]. Thus, undetectable RBM24 expression in neuronal tissues explains the greater extent of aberrant splicing of IKBKAP-FD exon 20 in neurons[25]. In this study, we revealed that SRSF6 is also involved in exon 20 recognition and that RECTAS specifically rectifies aberrant splicing by promoting the SRSF6 function.

## Results

**Binding of SRSF6 to an intronic splice enhancer (ISE) is required for IKBKAP-FD exon 20 inclusion.** IKBKAP-FD exon 20 has a suboptimal 5′ splice site, due to the IVS20 + 6 T > C mutation. It lacks complementarity to U1 snRNA at the −1 and +6 positions (Fig. 1a), and such suboptimal splice sites are more prone to be regulated by alternative splicing factors[11–13,26,27], as we reported for promotion of spliceosomal recognition of this 5′ splice site by RBM24 in skeletal muscle tissue[25]. The RNA sequencing (RNA-Seq) evaluation of the transcriptome of FD patient fibroblasts homozygous for the IKBKAP IVS20 + 6 T > C mutation following RECTAS treatment revealed a highly selective action of RECTAS on IKBKAP-FD exon 20, with the second-highest ΔPSI (percent spliced-in) of 618 altered splicing events with ≥0.1 of | ΔPSI | (Fig. 1b, c and Supplementary Data 1). As we supposed that RECTAS rescued exon 20 skipping as part of the overall context of splicing regulation, we checked whether RBM24 function is involved in RECTAS-induced exon 20 inclusion. Western blot for human tissues as well as HeLa and FD patient fibroblasts, responder cells to RECTAS[17], indicates RBM24 expression is exclusive in heart and skeletal muscle but absent in RECTAS responder cells, irrespective of RECTAS treatment (Fig. 1d). These observations ruled out the involvement of RBM24 in RECTAS-induced splice modification. Therefore, we hypothesized that RECTAS interacts with an unidentified splicing regulator(s) that promotes the restoration of normal exon 20 inclusion. To test this hypothesis, we examined loss-of-function study by RNAi knockdown for individual SRSFs, a major family of alternative splicing activators, and compared the extent of RECTAS-induced promotion of IKBKAP-FD exon 20 inclusion, using a dual fluorescent splicing reporter system we previously developed (Fig. 1e)[11,13,17]. We observed a marked reduction (~60%) in exon 20 inclusion when SRSF6 was knocked down (Fig. 1f, g), whereas depletion of nine other SRSFs yielded no or only a modest effect on this splicing event (Fig. 1f, g).

These observations indicate that SRSF6 is a crucial regulator of IKBKAP-FD exon 20 inclusion in the therapeutic splicing environment provided by RECTAS treatment. We further asked whether SRSF6 directly controls IKBKAP-FD exon 20 inclusion. For this purpose, we searched for potential SRSF6-recognition sites based on the systematic evolution of ligands by exponential enrichment (SELEX) motifs for SRSF6 with ESE finder 3.0[28,29] for a region downstream or upstream of the exon 20 5′-splice site (−100 to +100 nucleotides (nt)) of IKBKAP. We identified three candidate motifs 21, 45, and 58 nt downstream of the exon 20 5′ splice site (site-a, site-b, and site-c, respectively, in Fig. 1h). Subsequent validation of SRSF6 binding by RNA pull-down assay with biotin-conjugated RNAs revealed that only site-b has a high affinity to a cellular protein inferred to be SRSF6 (Fig. 1i). The band of expected SRSF6 disappeared in the pull-down products from SRSF6-knockdown cells, confirming its identity as SRSF6 (Fig. 1j). In addition, SRSF6 binding was lost when point mutations (site-b(mt) in Fig. 1h) were introduced in the SRSF6-binding motif (Fig. 1k). The functional significance of SRSF6 binding for IKBKAP-FD exon 20 recognition was subsequently examined by introducing a site-b(mt) mutation into the IKBKAP-

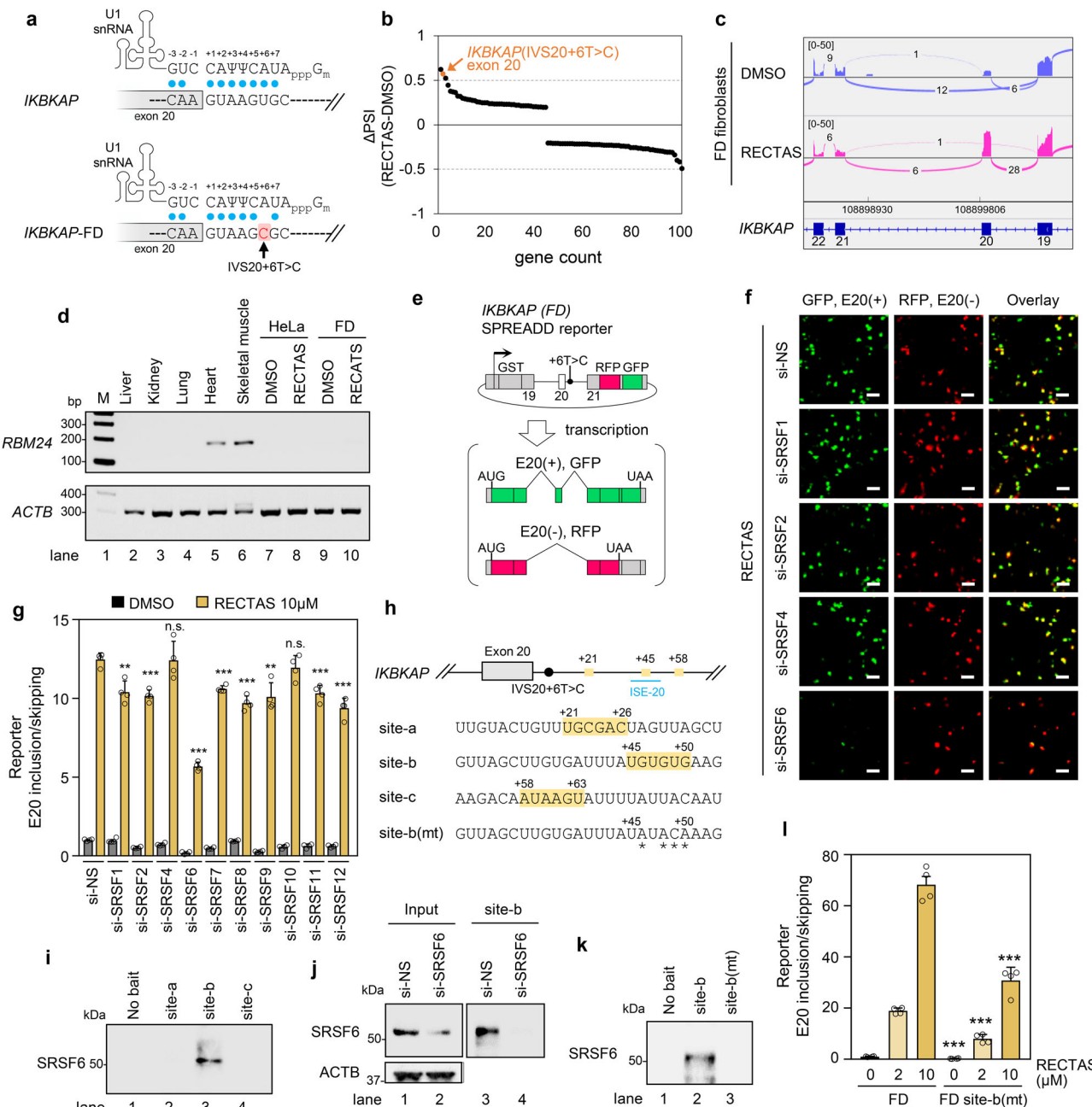

**Fig. 1 *IKBKAP*-FD exon 20 requires SRSF6 binding to an ISE. a** The *IKBKAP* exon 20 5′ splice site has nearly optimal pairing to U1 snRNA, whereas the IVS20 + 6 T > C creates an additional mismatch at +6 position, resulting in suboptimal recognition by U1 snRNP. Blue dots indicate base pairing. **b** Top 100 differential splice events by | ΔPSI (PSI$_{RECTAS}$-PSI$_{DMSO}$) | following RECTAS treatment (2 μM, 8 h) in FD patient fibroblasts. **c** Sashimi plot with junction read number of *IKBKAP*-FD exon 19–22 for DMSO- or RECTAS-treated FD patient fibroblasts. The displayed read count range was set to 0–50. **d** RT-PCR for *RBM24* expression in human tissues, and cell lines treated with 10 μM RECTAS or 0.1% DMSO for 24 h. *RBM24* and *ACTB* were detected with primers oAM669 + oAM671 and oAM13 + oAM14, respectively. **e** Diagram of reporter system for *IKBKAP*-FD exon 20 splicing[13,17]. GFP is translated when exon 20 is included (E20(+)), whereas RFP is translated when it is skipped (E20(−)). **f, g** Microscopic images (**f**) and quantification of *IKBKAP*-FD exon 20 inclusion (**g**), following treatment with RECTAS (10 μM) or 0.1% DMSO for 24 h in HeLa cells transfected with the indicated siRNAs for 72 h. **h** SRSF6-binding motifs starting at +21 nt (site-a), +45 nt (site-b), and +58 nt (site-c), predicted by ESE finder 3.0[28,29] are shown. RNA sequences for pull-down assays are indicated. Asterisks in site-b(mt) indicate mutated ribonucleotides. ISE-20 from Sinha et al.[30] is also indicated. **i–k** Western blot of SRSF6 by 1H4 antibody for RNA pull-down products or no bait RNA control for HeLa cells (**i**), site-b pull-down products for HeLa cells treated with siRNA for 72 h (**j**), and site-b and site-b(mt) RNA pull-down products for HeLa cells (**k**). **l** Quantification of *IKBKAP* exon 20 inclusion rate, following RECTAS treatment for 24 h in HeLa cells with *IKBKAP*-FD or site-b(mt)-mutant *IKBKAP* (FD site-b(mt)) reporter. Data from four replicates are shown in (**f**), (**g**), and (**l**), and two replicates are shown in (**i**), (**j**), and (**k**). Representative data of two experiments are shown in (**d**). Columns, mean; bars, SE; n.s., $P \geq 0.05$; **$P < 0.01$; ***$P < 0.001$ for unpaired two-tailed *t* test in (**g**) (vs. the RECTAS 10 μM of si-NS: si-SRSF1, $P = 2.3 \times 10^{-3}$; si-SRSF2, $P = 1.7 \times 10^{-4}$; si-SRSF4, $P = 0.94$; si-SRSF6, $1.5 \times 10^{-7}$; si-SRSF7, $P = 1.9 \times 10^{-4}$; si-SRSF8, $P = 1.0 \times 10^{-4}$; si-SRSF9, $3.0 \times 10^{-3}$; si-SRSF10, $P = 0.28$; si-SRSF11, $P = 6.1 \times 10^{-4}$; si-SRSF12, $P = 1.8 \times 10^{-4}$) and (**l**) (vs. the FD of corresponding RECTAS concentration: 0 μM, $P = 2.0 \times 10^{-4}$; 5 μM, $P = 4.4 \times 10^{-5}$; 10 μM, $P = 1.0 \times 10^{-4}$).

FD exon 20 splicing reporter, and a reduction in RECTAS-led exon 20 inclusion was observed, comparable to that seen in SRSF6 knockdown cells (~60%) (Fig. 1l). These combined observations indicate that site-b is an ISE, and whose recognition by SRSF6 is crucial to provide a splicing context that restores normal splicing of *IKBKAP*-FD exon 20 in the presence of RECTAS. We note that the identified SRSF6-dependent ISE overlaps with a 20-nt region with ISE activity (dubbed ISE-20) revealed by our previous antisense-oligonucleotide walk[30] (Fig. 1h). In addition, the SRSF6-binding site (+45 to +50) somewhat close to that of RBM24 (+13 to +29)[25] imply RBM24 binding may interfere with SRSF6 binding, and may impair the effect of RECTAS in tissues with RBM24 expression.

**Reciprocal control of *IKBKAP*-FD exon 20 splicing by REC-TAS and CDC-like kinase (CLK) inhibitors suggests that CLKs are pivotal regulators**. As SRSF6 is functionally activated by serine phosphorylation in its carboxyl-terminal RS domain, we next asked if kinases upstream of SRSF6 promote *IKBKAP*-FD exon 20 inclusion. For this purpose, we examined the effects of the following synthetic inhibitors of potential upstream SRSF kinases on *IKBKAP*-FD exon 20 recognition with our splicing reporter: TG003 for CLKs;[10,12,13] INDY, CaNDY, and ALGER-NON for dual-specificity tyrosine phosphorylation-regulated kinases (DYRKs)[31–33] and CLKs[11], and SRPIN340 for SRSF protein kinases (SRPKs)[34,35]. We used RECTAS and the plant cytokinin kinetin[17,36,37] as controls with opposite effect to that expected for inhibitors of SRSF6 kinases. Treatment of cells with TG003, INDY, CaNDY, and ALGERNON at 2 and 10 μM resulted in the dose-dependent promotion of *IKBKAP*-FD exon 20 skipping (Fig. 2a, b), in contrast to the increased inclusion of exon 20 by RECTAS treatment (Fig. 2a, b), whereas the exon-inclusion effect of kinetin was considerably weaker, consistent with our prior observation[17] (Fig. 2a, b). SRPIN340 had no effect on *IKBKAP*-FD exon 20 splicing (Fig. 2a, b). Some inhibitors of CLKs and DYRKs show cross-inhibition due to structural similarity at the ATP-binding pocket between these two kinase families[11,25,31–33,38,39], and depletion of CLK or DYRK isoforms by RNAi was conducted to distinguish their effects. As a result, we found that the depletion of CLK isoforms inhibited *IKBKAP*-FD exon 20 recognition similarly to SRSF6-depleted cells (Fig. 2c), whereas depletion of DYRK isoforms had no effect (Fig. 2c). Moreover, overexpression of CLK1 mimicked the effect of RECTAS on the inclusion of *IKBKAP*-FD exon 20 (Fig. 2d), whereas the kinase-inactive K191M mutant of CLK1 had no effect (Fig. 2d). Dependency of RECTAS activity on CLK activities was further assessed by evaluating RECTAS response in cells under CLK inhibition. We recently reported CaNDY as a potent pan-CLK inhibitor[11], and found that there is no exon 20 inclusion-promoting activity by RECTAS when CLKs were functionally inactivated (Fig. 2e). Moreover, we found SRSF6 overexpression also has exon 20 inclusion-promoting activity as seen in RECTAS treatment or CLK1 overexpression (DMSO conditions in Fig. 2f). However, this SRSF6 overexpression effect was also canceled when CLKs are inhibited (CaNDY conditions in Fig. 2f). These observations together support the idea that RECTAS depends on CLK1 activity for exon 20 inclusion-promoting activity. We further validated the effects of compound treatment (RECTAS, TG003, INDY, CaNDY, ALGERNON, and SRPIN340) and knockdown of SRSF6 or CLK in FD patient fibroblasts with the homozygous *IKBKAP*-FD IVS20 + 6 T > C mutation, and confirmed the increased *IKBKAP*-FD exon 20 skipping by CLK inhibition (Fig. 2g and Supplementary Fig. 1a) or depletion for SRSF6 or CLK (Fig. 2h and Supplementary Fig. 2a) for the endogenous transcripts. Primary fibroblasts without IVS20 + 6 T

> C mutation of *IKBKAP* showed no exon 20 skipping following CLK inhibition (Fig. 2i and Supplementary Fig 1b) or depletion for SRSF6 or CLK (Fig. 2j and Supplementary Fig. 2b), showing the effect of RECTAS depends on the suboptimal nature of exon 20 donor sites due to IVS20 + 6 T > C (Fig. 1a). In addition, we also compared RECTAS with compounds previously reported for exon 20 inclusion promotion[40], and we confirmed that RECTAS shows the highest activity (0.72 μM of $EC_{25}$) over those we tested (Fig. 2k, l). Two compounds, kinetin and (−)-epigallocatechin gallate (EGCG), reached $EC_{25}$ of RECTAS (Fig. 2k, m, n), but were 15.5- and 241-fold less potent and there was also cytotoxicity around the effective range for EGCG (Fig. 2n).

Having confirmed that CLK-dependent SRSF6 activation is required for restoration of normal splicing of *IKBKAP*-FD exon 20 by RECTAS, we then asked whether RECTAS affects the functions of CLKs and SRSF6. For this purpose, we synthesized a biotin-linked RECTAS analog (b-RECTAS) (Fig. 3a), which retained the *IKBKAP*-FD exon 20 inclusion-promoting activity, whereas biotin alone at 50 μM had no effect (Fig. 3b). We used b-RECTAS in a compound pull-down assay from HeLa cell lysates by neutravidin to evaluate its binding affinity to CLK1 and SRSF6. We found that b-RECTAS binds to CLK1 in a dose-dependent manner but not to SRSF6 (Fig. 3c). Recombinant CLK1 protein also bound to b-RECTAS in a dose-dependent manner, indicating their direct interaction (Fig. 3d). The interaction between b-RECTAS and CLK1 was effectively competed out by incubating with 500 nmol of free RECTAS (lane 6 in Fig. 3d). Western blotting for RECTAS-treated FD patient fibroblasts revealed that the SRSF6 phosphorylation was enhanced following RECTAS treatment (Fig. 3e), and this effect was detected by 6 h and disappeared by 10 h after washout of RECTAS (Fig. 3f). Since it was recently reported that several small molecules stabilize U1 snRNA-donor paring for *SMN* exon 7 through 5′-splice-site bulge repair[8], we tested whether RECTAS has such activity on *IKBKAP* exon 20 donor site. RNA pull-down assay by *IKBKAP* exon 20 donor sequences with or without IVS20 + 6 T > C reproduced weakened affinity to U1 snRNP (detected by U1-70k and SmB/B′ subunits) by IVS20 + 6 T > C mutation (lanes 4 and 6 in Fig. 3g), whereas RECTAS had no effect on their affinity, denying its role in donor-U1 snRNA paring (lanes 5 and 7 in Fig. 3g).

**The therapeutic effect of RECTAS in FD disease models**. Having established the mechanism and target selectivity of RECTAS, we then assessed its therapeutic potential for FD, in two FD disease models: iPSC-SNs and *IKBKAP*-FD transgenic mice. As alternative splicing undergoes tissue-dependent regulation due to a varied distribution of RNA-binding proteins[41], it is desirable to assess therapeutic potential in the context of sensory or autonomic neurons, which are affected in FD patients[18–20]. For efficient preparation of induced pluripotent stem cells-derived sensory neurons (SNs), we employed the embryoid-body method for the initial differentiation step (detailed in "Methods"). This modification resulted in higher viability and rapid derivation of the SNs (BRN3A + /TUBB3 + ) on day 12 (Fig. 4a) (see "Methods" for details). The effect of the compounds was tested by adding kinetin, RECTAS, or solvent only (0.1% dimethyl sulfoxide (DMSO)) to the culture media during the whole differentiation process. We found that 10 μM of RECTAS completely restored *IKBKAP*-FD exon 20 inclusion (lane 7 in Fig. 4b), while 10 μM of kinetin did not eliminate exon 20 skipping completely in patient iPSC-SNs (lane 6 in Fig. 4b). The effect of RECTAS on exon 20 inclusion was transient, and beginning around 2 h after the incubation and lasting for ~4 h after washout (Supplementary Fig. 3). We also examined the effect of CLK inhibition for RECTAS responsivity in iPSC-SNs. Consistent with observations

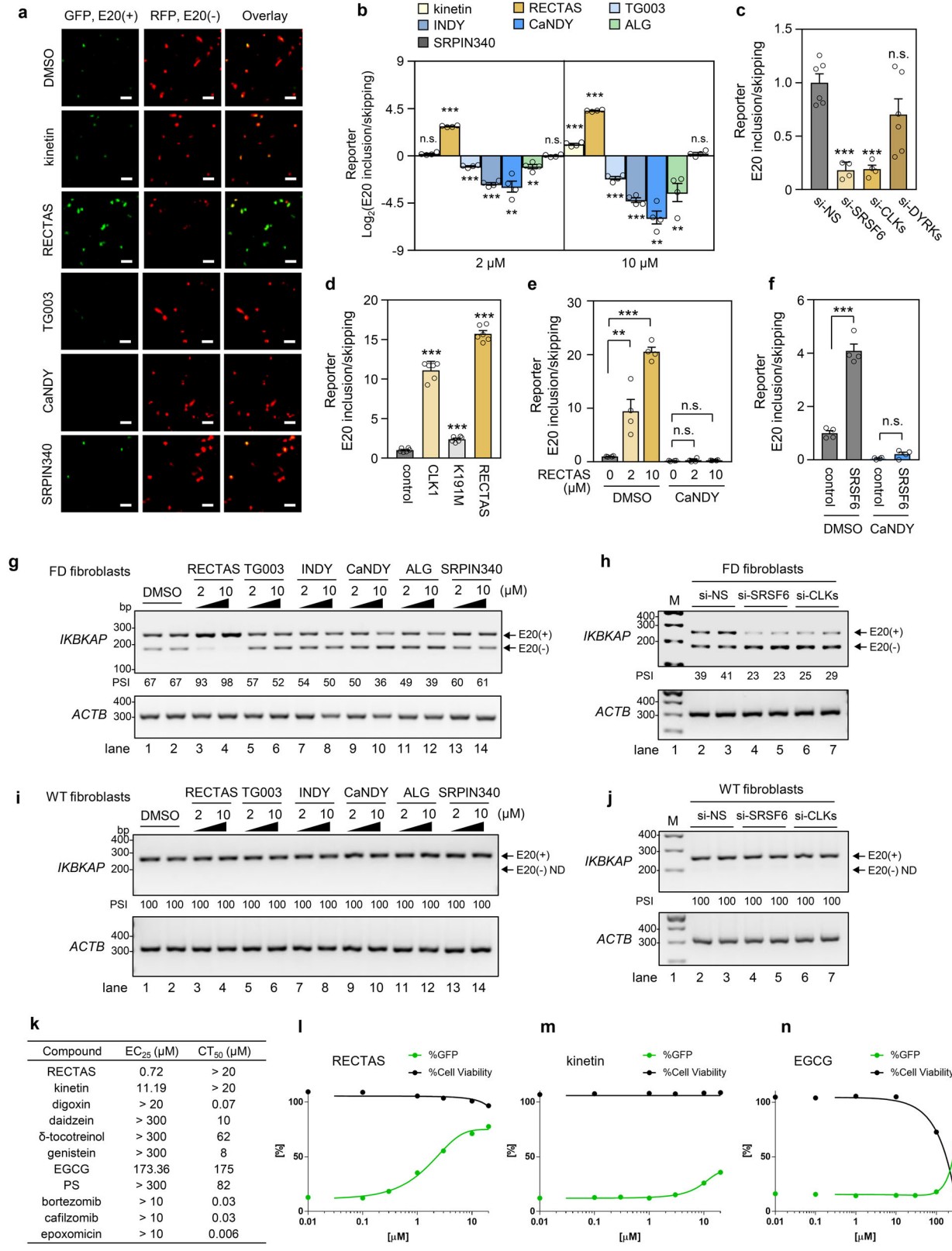

in *IKBKAP*-FD reporter-transfected HeLa cells, iPSC-SNs showed impaired RECTAS activity by CLK inhibition with CaNDY treatment (Fig. 4c).

Next, we attempted to test the therapeutic potency of RECTAS in an FD mouse model. Prior to in vivo administration, we examined whether RECTAS also has an exon 20 inclusion-promotive effect in mice neural tissue culture, neuro 2A cells. We

confirmed a marked induction of exon 20 inclusion in neuro 2A (Fig. 4d, e), indicating activity of RECTAS for human *IKBKAP*-FD exon 20 is retained in murine cells. RT-PCR was also confirmed for RECTAS-induced exon 20 inclusion for human *IKBKAP*-FD, without alteration in endogenous mouse *Ikbkap* exon 19 (Fig. 4f), which is homologous to exon 20 in human and lacking IVS20 + 6 T > C mutation (Supplementary Fig. 4). Then

**Fig. 2 A pivotal control of *IKBKAP*-FD exon 20 splicing by RECTAS and a CLK inhibitor. a, b** Microscopic images (**a**) and *IKBKAP*-FD exon 20 inclusion rates (**b**) following compound treatment (2 or 10 μM) or solvent only (0.1% DMSO) for 24 h in HeLa cells transfected with *IKBKAP*-FD reporter. ALG indicates ALGERNON in (**b**). **c** HeLa cells with *IKBKAP*-FD reporter were transfected with siRNA for si-SRSF6, si-CLKs, si-DYRKs, or si-NS for 72 h, and *IKBKAP*-FD exon 20 inclusion rates were plotted by GFP/RFP ratio. **d** *IKBKAP*-FD exon 20 inclusion was quantified by the GFP/RFP ratio of *IKBKAP*-FD reporter in HeLa cells, transfected with empty vector (control), CLK1 expression vector, or inactive mutant CLK1 (K191M) expression vector for 24 h, or treated with RECTAS (2 μM) for 24 h. **e, f** *IKBKAP*-FD exon 20 inclusion rates in response to RECTAS treatment for 24 h (**e**) or transfection with SRSF6 expression vector for 24 h (**f**) were quantified by GFP/RFP ratio in HeLa cells, transfected with *IKBKAP*-FD reporter with or without CaNDY treatment (10 μM, 24 h). control, empty p3XFLAG-CMV14 vector transfection in (**f**). **g, h** RT-PCR for *IKBKAP*-FD exon 19–21 for FD patient fibroblasts (P2) treated with compounds (2 or 10 μM) or solvent only (0.1 % DMSO) for 24 h (**g**), or transfected with si-NS, si-SRSF6, or si-CLKs for 72 h (**h**). **i, j** RT-PCR for *IKBKAP*-FD exon 19–21 for fibroblasts from a healthy individual (C1) treated with compounds (2 or 10 μM) or solvent only (0.1% DMSO) for 24 h (**i**), or transfected with si-NS, si-SRSF6, or si-CLKs for 72 h (**j**). **k–n** 25% effective concentration (EC$_{25}$) and 50% cytotoxic concentration (CT$_{50}$ for 72 h treatment) were shown for indicated compounds. EC$_{25}$ was determined with %GFP of 20 μM RECTAS set to 100%. Individual plots for RECTAS (**l**), kinetin (**m**), and (−)-epigallocatechin gallate (EGCG) (**n**) are shown separately. Data from four replicates are shown in (**a**), (**b**), (**c**), (**e**), and (**f**), six replicates in (**d**), four replicates for %GFP, and three replicates for %Cell viability in (**k–n**). Representative data from two replicates are shown in (**g–j**). Columns, mean; bars, SE; n.s., $P \geq 0.05$; *$P < 0.05$; **$P < 0.01$; ***$P < 0.001$ for unpaired two-tailed $t$ test in (**b**) (vs. the 0.1% DMSO control) (kinetin: 2 μM, $P = 9.2 \times 10^{-2}$; 10 μM, $P = 8.8 \times 10^{-5}$; RECTAS: 2 μM, $P = 3.2 \times 10^{-8}$; 10 μM, $3.2 \times 10^{-9}$; TG003: 2 μM, $2.3 \times 10^{-5}$; 10 μM, $1.3 \times 10^{-5}$; INDY: 2 μM, $1.5 \times 10^{-6}$; 10 μM, $1.6 \times 10^{-6}$; ALG: 2 μM, $P = 4.1 \times 10^{-3}$; 10 μM, $P = 7.1 \times 10^{-3}$; SRPIN340: 2 μM, $P = 0.90$; 10 μM, $P = 0.27$), (**c**) (vs. si-NS: si-SRSF6, $P = 6.7 \times 10^{-5}$; si-CLKs, $P = 7.1 \times 10^{-5}$; si-DYRKs, $P = 0.11$), (**d**) (vs. control vector: CLK1, $P = 1.1 \times 10^{-9}$; K191M, $P = 1.0 \times 10^{-5}$; RECTAS, $P = 6.1 \times 10^{-12}$), (**e**) ($P = 9.0 \times 10^{-3}$ and $P = 3.9 \times 10^{-7}$ for RECTAS 2 μM and 10 μM without CaNDY, and $P = 0.53$ and 0.13 for RECTAS 2 μM and 10 μM with CaNDY), and (**f**) (DMSO, $P = 3.1 \times 10^{-5}$; CaNDY, $P = 8.0 \times 10^{-2}$). E20 (+), exon 20 inclusion product; E20 (−), exon 20 skipping product in (**a**), (**g**), (**h**), (**i**), and (**j**); ND, not detectable in (**i**) and (**j**); PSI, percent spliced-in in (**g–j**). *IKBKAP* and *ACTB* were detected with primers oAM138 + oAM139 and oAM13 + oAM14, respectively in (**g–j**).

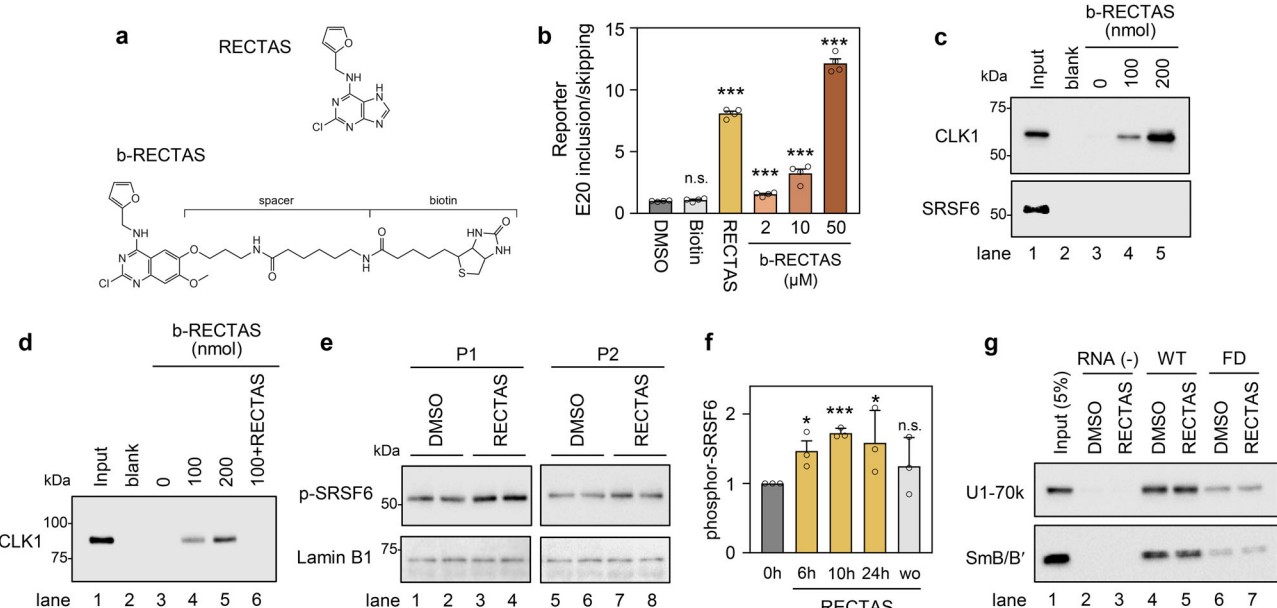

**Fig. 3 RECTAS directly interacts with CDC-like kinase 1 (CLK1), and it enhances cellular SRSF6 activity. a** Structures of RECTAS and b-RECTAS. **b** Extent of *IKBKAP*-FD exon 20 inclusion represented by GFP/RFP ratio, following treatment with biotin (50 μM), RECTAS (2 μM), b-RECTAS (2, 10, and 50 μM), or solvent only (0.1% DMSO) for 24 h in reporter-transfected HeLa cells. **c, d** Western blotting of pull-down products by b-RECTAS (0, 100, or 200 nmol/ reaction) for HeLa cells transfected with a flag-tagged CLK1 expression vector (**c**) or GST-tagged recombinant CLK1 (**d**). RECTAS (500 nmol) was incubated with 100 nmol b-RECTAS for the competition assay in (**d**). **e, f** Western blot (**e**) for phosphor SRSF6 in FD patient primary fibroblasts (P1 and P2), treated with RECTAS (10 μM) or solvent only (0.1% DMSO) for 24 h. **f** Phosphor SRSF6 in FD fibroblasts (P1) was quantified at 0, 6, 10, 24 h after RECTAS (10 μM) treatment (columns, 0 h, 6 h, 10 h, and 24 h), as well as 10 h after washout of compound following RECTAS treatment (10 μM) for 24 h (column, wo). Lamin B1 served as a loading control in (**e**). **g** Western blot for U1-70k and SmB/B′ in pull-down products with biotin-conjugated RNA surrounding IKBKAP exon 20 donor site with (oAM154) or without (oAM153) IVS20 + 6 T > C mutation (labeled as FD and WT, respectively). Pull-down was conducted in the presence of 10 μM RECTAS with 1% DMSO, or 1% DMSO only. RNA (−), control sample without RNA bait. Input, HeLa nuclear extract of 5% input amount. Representative data from four replicates were shown in (**b**). Representative data from two replicates were shown in (**c**), (**d**), and (**g**). Data from two experiments are shown in (**e, f**) and three experiments are shown in (**f**). Columns, mean; bars, SE; n.d. $\geq 0.05$; *$P < 0.05$; ***$P < 0.001$ for unpaired two-tailed $t$ test in (**b**) (biotin, $P = 0.26$; RECTAS, $P = 2.5 \times 10^{-8}$; b-RECTAS 2 μM, $5.5 \times 10^{-4}$; b-RECTAS 10 μM, $7.5 \times 10^{-4}$; b-RECTAS 50 μM, $9.6 \times 10^{-8}$) and one-tailed $t$ test in (**f**) (vs. the 0 h control: 6 h, $P = 1.6 \times 10^{-2}$; 10 h, $P = 2.8 \times 10^{-5}$; 24 h, $P = 4.8 \times 10^{-2}$; wo, $P = 0.18$). No sample was loaded in blank lanes in (**c, d**).

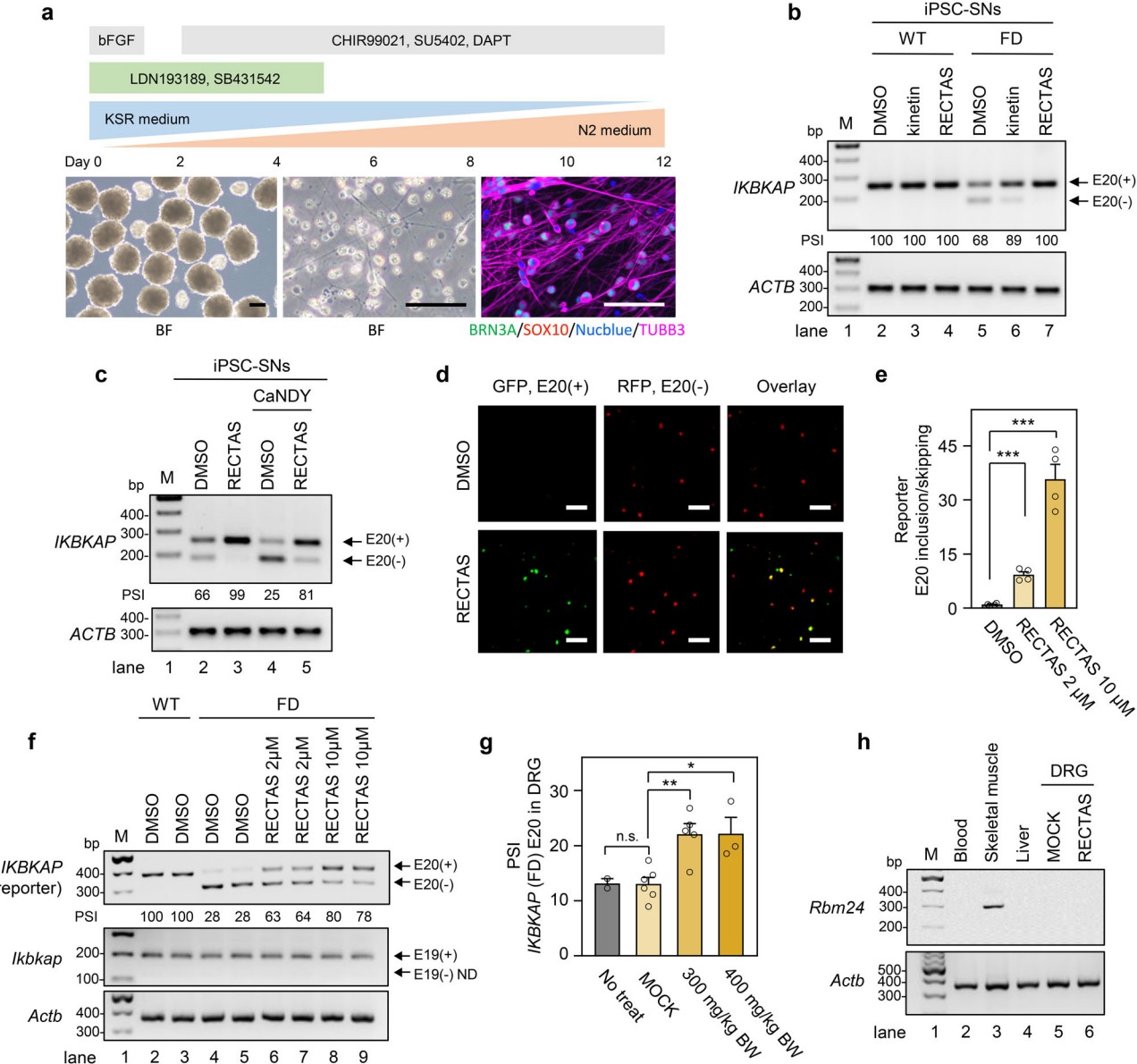

**Fig. 4 RECTAS restores *IKBKAP*-familial dysautonomia (FD) exon 20 inclusion in patient induced pluripotent stem cells-derived sensory neurons (iPSC-SNs) and transgenic mice. a** Diagram for preparation of iPSC-SNs, showing time-course (upper), microscopic images of spheroids at day 2 (lower, left), and differentiated iPSC-SNs on day 5 (lower middle for brightfield and lower right for BRN3A/SOX10/Nucblue/TUBB3 immunocytostaining image). BF, bright field image; scale bars, 100 μm. **b**, **c** RT-PCR for *IKBKAP* exon 19–21 and *ACTB* as a loading control for differentiated iPSC-SNs from a healthy donor (WT) or FD patient (FD) on day 12 (**b**), and iPSC-SNs (day 12) treated with CaNDY (10 μM) or 0.1% DMSO for 2 h, followed by addition of RECTAS (10 μM) or 0.1% DMSO in media and subsequent incubation for 22 h (**c**). *IKBKAP* and *ACTB* were detected with primers oAM138 + oAM139 and oAM13 + oAM14, respectively. **d**, **e** Microscopic images (**d**) and quantification for the extent of *IKBKAP*-FD exon 20 inclusion (**e**), following treatment with RECTAS (10 μM for (**d**), and 2 and 10 μM for (**e**)) or solvent only (0.1% DMSO) for 24 h in neuro 2A cells transfected with the *IKBKAP*-FD reporter. Bars, 100 μm in (**d**). **f** RT-PCR for exogenous human *IKBKAP* (oAM124 + oAM126, primers designed for vector backbone) and endogenous mouse *Ikbkap* (oAM666 + oAM667, primers designed for exon 18 and 20) in neuro 2A cells transfected with *IKBKAP*-WT or -FD reporter and treated with 2 or 10 μM RECTAS or 0.1% DMSO for 24 h. *Actb* (oAM364 + oAM366) served as a loading control. **g** Quantification of PSI for *IKBKAP*-FD exon 20 in DRG from *IKBKAP*-FD-humanized transgenic mice administered RECTAS (300, or 400 mg/kg BW, p.o.) with a booster administration of the same dose at 4 h, and DRG were collected 8 h after the first administration to be applied for RT-PCR for *IKBKAP* exon 19–21 with primer set of HsIKAPRT18F + HsIKAPRT21R ($n = 2$ for no treatment, $n = 6$ for MOCK, $n = 5$ for 300 mg/kg BW RECTAS, and $n = 3$ for 400 mg/kg BW RECTAS). MOCK, control for the vehicle only. **h** RT-PCR for *Rbm24* in blood cells, skeletal muscle, and liver tissues from 2-month-old B6 mouse, as well as DRG from MOCK or RECTAS (300 mg/kg BW)-administered *IKBKAP*-FD transgenic mice. *Actb* served as a loading control. *Rbm24* and *Actb* were detected by primers Rbm24-F + Rbm24-R and oAM365 + oAM366, respectively. Data from three iPSC clones are shown in (**a–c**), four replicates were shown in (**d–e**), two replicates are shown in (**f**). Representative data from two experiments are shown in (**h**). E20 (+), exon 20 inclusion product; E20 (−), exon 20 skipping product; E19 (+), exon 19 inclusion product; E19 (−), exon 19 skipping product in (**b**), (**c**), and (**f**). Columns, mean; bars, SE; n.s., $P \geq 0.05$; **$P < 0.01$; ***$P < 0.001$ for unpaired two-tailed $t$ test in (**e**) (vs. DMSO control: 2 μM, $P = 5.6 \times 10^{-5}$; 10 μM, $P = 1.5 \times 10^{-4}$) and (**g**) (vs. MOCK control: no treat, $P = 0.97$; 300 mg/kg BW, $2.4 \times 10^{-3}$; 400 mg/kg BW, $1.0 \times 10^{-2}$); PSI, percent spliced-in in (**b**), (**c**), and (**f**); ND, not detectable in (**f**).

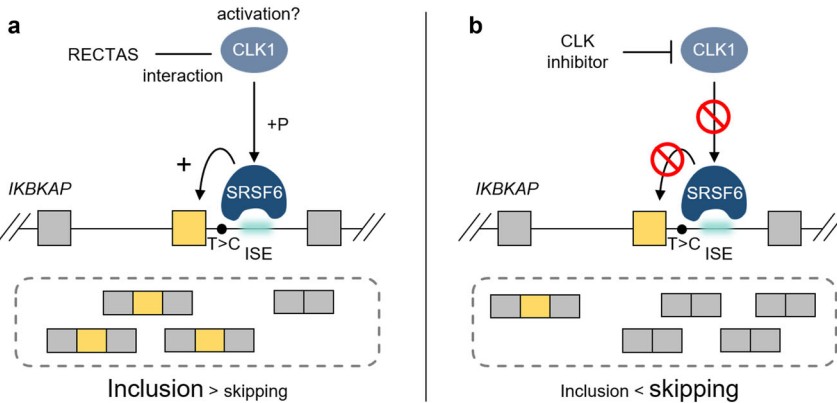

**Fig. 5 Diagram for bidirectional control of *IKBKAP*-FD exon 20 by small molecules. a** SRSF6 binding at the ISE is a major factor in the therapeutic splicing environment induced by RECTAS treatment. The requirement of CLK1 for this process and the direct interaction between CLK1 and RECTAS implies a steric effect for the activation of cellular CLK1. **b** For its dependence on CLK1 and SRSF6 for recognition, *IKBKAP*-FD exon 20 splicing can be controlled in a reverse direction by CLK inhibitor treatment.

we measured the effect of per os (p.o.) administration of RECTAS, using previously established *IKBKAP*-FD transgenic mice, in which an IVS20 + 6 T > C-mutant human *IKBKAP* gene was introduced in a BAC clone, in the presence of the endogenous murine wild-type *Ikbkap*, as an asymptomatic splicing model[42]. RECTAS was orally administered at 300 or 400 mg/kg body weight (BW) with a booster administration at 4 h intervals. Eight hours after the first administration, total RNA was extracted from dorsal root ganglia (DRG) to assess the splicing profile in sensory neurons. RT-PCR revealed enhanced inclusion (about 70% increase in PSI from ~12.5 to ~22%) of *IKBKAP*-FD exon 20 following RECTAS administration at both doses in DRG (Fig. 4g). We subsequently conducted transcriptome analysis for DRG, excised from MOCK and RECTAS-treated (300 mg/kg BW) groups. We found *IKBKAP*-FD exon 20 inclusion among the highly promoted exons (13th highest alterations) (Supplementary Fig. 5 and Supplementary Data 2), consistent with its selective effect in patient primary fibroblasts (Fig. 1b, c and Supplementary Data 1). Moreover, we confirmed there was no *Rbm24* expression detectable in DRG irrespective of RECTAS administration (Fig. 4h), ruling out the involvement of *Rbm24* as shown for human tissue cultures (Fig. 1d). Finally, we assessed the preclinical profile of RECTAS, checking for effects on body weight change and food consumption, as well as conducting necropsy and hematology tests, both after single p.o. administration (0, 100, 300, 1000 mg/kg BW) and 14-day repetitive p.o. administration (0, 40, 200, and 1000 mg/kg BW/day) (Supplementary Table 1). There was no genotoxicity for RECTAS in the micronucleus and Ames tests (Supplementary Table 1). These preclinical data together indicate that there was no anticipated adverse effect associated with RECTAS administration, assumingly due to its high target selectivity for *IKBKAP*-FD exon 20 (Fig. 1b, c and Supplementary Fig. 5).

## Discussion

Recent studies revealed that the prevalence of splicing mutations is much higher than previously thought, as ~30–50% of disease-associated mutations are estimated to cause abnormal splicing[43,44], and the total number is further increasing due to the identification of deep-intronic splicing mutations through recent whole-genome sequencing and transcriptome analysis[13,45–48]. Therefore, an innovative therapeutic strategy to target mis-regulation of splicing is urgently needed. Small-molecule compounds that target *trans*-acting splice factors offer an opportunity

to systematically target a particular set of mis-splicing events that are regulated by a similar splicing context. In this study, we identified alternative exons that show a dependency on a subset of SRSFs and CLK kinases are rational targets for selective manipulation in both promotive and suppressive ways (Fig. 5). Our data indicate that enhanced SRSF6 activity is a crucial aspect of the splicing environment induced by RECTAS treatment, and the requirement of CLKs in this process suggests that the direct interaction of RECTAS with CLK1 (Fig. 3c, d) sterically activates cellular CLK1. As a precedent for PDK1, the regulatory kinase of AGC kinases, there is an allosteric small-molecule activator that binds to the HM/PIF-pocket to induce a conformational change[49]. We assume that RECTAS binding modifies the intra-molecular conformation of CLK1 to promote its catalytic activity or provide accessibility of a signal mediator upstream of CLK1. Understanding the binding mode of RECTAS to CLK1 and the activating pathway for CLKs in the context of suboptimal donor RNA recognition by SRSF6 will provide further insights into this regulation.

We found that RECTAS p.o. administration of 200–300 mg/kg BW for 8 h achieved ~22% PSI for exon 20 in DRG of *IKBKAP*-FD transgenic mice, versus ~12.5% baseline PSI in mock or untreated controls (Fig. 4g). In the recent report from Morini et al.[50], as little as ~12.5% of *IKBKAP* exon 20 PSI (versus ~10% baseline) by kinetin administration (200 mg/kg BW/day in the diet for 6 months) was sufficient for the phenotypic rescue of sensory loss. Though these data were obtained with different transgenic mice and cannot be directly compared, we expect that RECTAS should exhibit higher therapeutic efficacy in vivo than kinetin, which already completed a phase 1 clinical trial in FD patients (NCT02274051 in ClinicalTrials.gov). Kinetin has been reported to have high target selectivity in FD patient fibroblasts, and our study revealed a similar target-selective profile for RECTAS. Other RECTAS-responsive exons, such as *TATA-box binding protein associated factor, RNA polymerase I subunit A* (*TAF1A*) exon 4 and *par-3 family cell polarity regulator* (*PARD3*) exon 21 (Supplementary Data 1) had no apparent effect on cell viability. Presumably, this is because alternative splicing occurs physiologically for most genes, and the safety profile determined in preclinical tests further confirmed that RECTAS is well-tolerated in rodents. In addition, we confirmed the higher activity of RECTAS than previously reported *IKBKAP* splice modifiers in a reporter assay (Fig. 2k), which further highlights the efficient activity of RECTAS for *IKBKAP*-FD exon 20 inclusion. These observations together suggest that RECTAS may prove to be a

potent agent for the treatment of FD, and ongoing medicinal chemistry indicates further improvement in potency.

Searching in existing databases, there are numerous splicing mutations that resemble the context of *IKBKAP*-FD exon 20. For example, a splicing mutation in the *potassium voltage-gated channel subfamily Q member 1* (*KCNQ1*), which is responsible for congenital long QT syndrome type I, has a similar splice enhancer motif, and the impaired exon of *KCNQ1* is also manipulated bidirectionally in a similar manner as *IKBKAP* IVS20 + 6 T > C (our unpublished observations). The opportunity for bidirectional regulation by CLK inhibitors and agonists offers a strategic approach for individual splicing contexts, as promoting recognition of a target exon, on the one hand, provides a therapeutic effect for splicing mutations like the IVS20 + 6 T > C mutation of *IKBKAP*, whereas inhibition of a target exon, on the other hand, offers a therapeutic opportunity for splice-site-creating splicing mutations, as we previously demonstrated the therapeutic effects of CLK inhibitors in EDA-ID with the IVS6 + 866 C > T mutation of *NEMO*[13] and cystic fibrosis with the c.3849 + 10 kb C > T mutation of *CFTR*[11]. CLK-targeting strategies may also become increasingly relevant, as recent whole-genome and transcriptome analyses highlighted the widespread presence of deep-intronic mutations that lead to the creation of cryptic pseudoexons[13,45–48].

In addition, numerous splicing mutations have also been identified in recent cancer genomics studies[51,52], although their individual contributions to malignancy remain largely unknown. CLK inhibitors (e.g., T-025[53] and SM08502[54]) and SRPK inhibitors (e.g., SPHYNX[55,56], SRPIN340[35], and SRPKIN-1[57]) described by other groups and us were recently reported to have anti-cancer activity in several cancer types. Moreover, small-molecule compounds that target the essential splice factor SF3B, such as spliceostatin A and E7107[58,59], have shown efficacy in leukemia models harboring mutations in the same or different *trans*-acting splice factors, namely *SF3B1*, *SRSF2*, or *U2AF35*, which sensitize the cells to the inhibitors[60–63], though inhibition of SF3B also evokes global intron retention and cytotoxicity. In this regard, targeting CLKs by RECTAS, with a target selectivity sufficient for a favorable safety profile in preclinical tests, may provide an alternative therapeutic strategy to target cancer cells with vulnerable splicing machinery or CLK-dependent causal splicing misregulation.

Overall, our current observations with *IKBKAP*-FD exon 20 not only highlight RECTAS as a potential therapeutic agent for FD but also suggest the feasibility of chemical control of disease-associated missplicing by manipulating the splicing environment composed of SRSFs and CLKs in target cells. Facing the increasing prevalence of splicing mutations associated with human diseases, mechanism-oriented precision medicine for missplicing, as exemplified by this study, may provide a new framework for targeting such splicing diseases.

## Methods

**Cell lines.** HeLa cells were obtained from the Japanese Collection of Research Bioresources (JCRB) Cell Bank of the National Institutes of Biomedical Innovation, Health and Nutrition (NIBIOHN) (JCRB9004) (Osaka, Japan) and cultured in Dulbecco's modified Eagle medium (DMEM) (Bibco of Thermo Fisher Scientific, Waltham, MA, USA) supplemented with 10% fetal bovine serum, 100 U/mL penicillin, and 100 μg/mL streptomycin. Neuro 2A cells were obtained from JCRB Cell Bank of NIBIOHN (IFO50081), and cultured in DMEM (GIBCO of Thermo Fisher Scientific, Waltham, MA, USA) supplemented with 1× GlutaMAXI (GIBCO of Thermo Fisher Scientific), 10% fetal bovine serum, 100 U/mL penicillin, and 100 μg/mL streptomycin. Fibroblasts from 19- and 26-year-old male patients with FD were provided by Coriell Institute (Camden, NJ, USA) (GM02342 (P1) and GM00850 (P2), respectively) and cultured in DMEM supplemented with 15% fetal bovine serum, 100 U/mL penicillin, and 100 μg/mL streptomycin. Primary fibroblasts from healthy donors (36-year-old male (TIG-114, C1) and 40-year-old female (TIG-108, C2)) were obtained from JCRB Cell Bank of NIBIOHN, and cultured in DMEM supplemented with 10% fetal bovine serum, 100 U/mL penicillin, and 100 μg/mL streptomycin. All cells were maintained in a cell incubator at

37 °C with 5% $CO_2$. The tissue culture study was conducted under ethical approval by the ethics committee of Kyoto University Graduate School and Faculty of Medicine.

**iPSCs.** Control- and patient-derived iPSCs C1 and M1 were established previously[24,64]. iPSCs were maintained on Matrigel (Corning, #354277) as colonies in mTeSR1 medium (STEMCELL Technologies, #85850), according to the manufacturer's protocol. iPSC-SNs were prepared following a published protocol[24] with several modifications; the undifferentiated iPSCs were trypsinized to single cells and re-aggregated on AggreWell 800 plates (STEMCELL Technologies, 34815) in Primate embryonic stem cell medium (REPROCELL, RCHEMD001) with bFGF (5 ng/mL, WAKO, 062-06661) and Y-27632 (10 μM, 036-24023). On day 5, spheroids were transferred to Matrigel-coated cell culture plates and differentiated further. The medium was gradually changed to DMEM/F12 supplemented with N2 supplement, nonessential amino acids, and Glutamax supplement in the presence of LDN-193189 0.1 μM and SB431542 10 μM (Repro-nLS, Cellagen Technology, K6570-500) for days 0–4 and CHIR99021 3 μM, DAPT 10 μM, and SU5402 10 μM (n3i, Cellagen Technology, K6571-500) for days 2–11. On day 12, the medium was changed to a 1:1 mixture of the above N2 medium and Neurobasal medium supplemented with B27 supplement, and with brain-derived neurotrophic factor 10 ng/mL, glia-derived neurotrophic factor 10 ng/mL, and dbcAMP 1 mM for maturation. For immunocytochemistry, cells were fixed with 4% paraformaldehyde for 15 min, permeabilized with 0.1% Triton X-100 in D-PBS (−) for 15 min, and blocked with 3% bovine serum albumin (BSA) in D-PBS (−) for 1 h. Then the cells were incubated with primary antibodies in 3% BSA in D-PBS (−) for 2 h, followed by visualization with appropriate secondary antibodies. The nuclei were counterstained with Hoechst 33342 (Thermo Fisher Scientific). Experiments using iPSCs were conducted under ethical approval by the ethics committee of Kyoto University Graduate School and Faculty of Medicine.

**Antibodies.** The following antibodies were used for Western blotting: anti-U1-70k mouse monoclonal antibody (9C4.1) (05-1588, Merk Millipore, Burlington, MA) at 1:500; anti-SmB/B′ mouse monoclonal antibody (Y12) (MA5-13449, Thermo Fisher Scientific) at 1:500; anti-CLK1 rabbit polyclonal antibody (ARP52021_P050, Aviva systems biology, San Diego, CA) at 1:500; anti-CLK2 rabbit polyclonal antibody (ab65082, Abcam, Cambridge, UK) at 1:500; anti-CLK3 rabbit polyclonal antibody (3256, Cell Signaling Technology, Danvers, MA) at 1:500; anti-CLK4 rabbit polyclonal antibody (ab104321, Abcam) at 1:500; anti-β-actin mouse monoclonal antibody (Ac-15) (sc-69879, Santa Cruz Biotechnology, Dallas, TX) at 1:4000; anti-SR protein (1H4G7) mouse monoclonal antibody (33–9400, Thermo Fisher Scientific) at 1:200 for phosphorylated SR proteins; anti-Lamin B1 (EPR8985) rabbit monoclonal antibody (ab133741, Abcam, Cambridge, UK) at 1:500. The following antibodies were used for immunocytostaining: anti-OCT4 rabbit monoclonal antibody (T.631.9) (MA5-14845, Thermo Fisher Scientific) at 1:400, anti-SSEA4 mouse monoclonal antibody (MC-813-70) (MA1-021, Thermo Fisher Scientific) at 1:500, anti-SOX10 rabbit monoclonal antibody (EPR4007) (ab155279, Abcam) at 1:250, anti-BRN3A mouse monoclonal antibody (5A3.2) (MAB1585, Merck Millipore) 1:200, and anti-beta III tubulin (TUBB3) rabbit monoclonal antibody (EP1569Y) (ab52623, Abcam) 1:500. Counterstaining was done with NucBlue cell stain (Thermo Fisher Scientific).

**Compounds.** Kinetin was purchased from Nacalai Tesque (Tokyo, Japan). (−)-epigallocatechin gallate (EGCG), daidzein, genistein, epoxomicin, and bortezomib were purchased from FUJIFILM Wako Pure Chemical (Osaka, Japan). δ-tocotrienol was produced by LKT Laboratories, Inc. (St. Paul, MN). Phosphatidylserine was produced by Larodan Fine Chemicals AB (Solna, Sweden). Digoxin was produced by Cayman Chemical (Ann Arbor, MI). Carfilzomib was purchased from Cell Signaling Technology. b-RECTAS was synthesized in-house from RECTAS. Compound stock solutions were prepared with DMSO, except for phosphatidylserine, which was dissolved in chloroform:methanol solution (95:5 % volume).

**IKBKAP (IVS20+6T>C) humanized mice.** The *IKBKAP* (IVS20 + 6 T > C) transgenic mouse strain used in this study was obtained from Drs. Pickel and Slaugenhaupt[42]. Mice were bred in the light/dark cycle of 12 h/12 h, 40–60% humidity and 69–72 °F temperature. RECTAS in suspension with 0.5% carboxymethyl cellulose was administered to more than 8-week-old mice p.o. at a dose of 300 or 400 mg/kg BW. Four hours later, an equivalent booster dose was given, and after another 4 h, the mice were euthanized and RNA extraction from dorsal root ganglia was performed. Extracted RNA was further analyzed by RT-PCR to measure the PSI value for exon 20 inclusion of *IKBKAP*. An animal study using *IKBKAP* (IVS20 + 6 T > C) mice was conducted under ethical approval by Cold Spring Harbor Laboratories.

**Reverse transcription-coupled polymerase chain reaction (RT-PCR).** Total RNA purification was conducted by direct-zol RNA miniprep (zymo research, Irvine, CA) or RNeasy mini kit (Hilden, Germany) from a cell line or homogenized tissue block of C57BL/6 mouse in TRIzol (Thermo Fisher Scientific). Blood total RNA from C57BL/6 mouse was extracted by NucleoSpin RNA blood column

(Macherey-Nagel, Dueren, Germany), followed by addition of TRIzol (Thermo Fisher Scientific) and another round of purification by direct-zol RNA miniprep (zymo research). Total RNAs from human liver (No. 636531), kidney (No. 636529), lung (No. 636524), heart (No. 636532), and skeletal muscle (No. 636534) were purchased from Takara Bio (Shiga, Japan). For RT-PCR, total RNA was applied for reverse transcription by PrimeStar Reverse Transcriptase (Takara Bio) with random hexamers, and the resulting products were amplified with ExTaq DNA polymerase (Takara Bio) with target-specific primer sets. Primers used for RT-PCR are listed in Supplementary Table 2. Detection was conducted with ChemiDoc MP Imaging System (Bio-Rad, Hercules, CA) and analyzed by Image Lab software (6.0.1) (Bio-Rad).

**RNAi**. RNAi was conducted by lipofection of small interfering RNA (siRNA) (Silencer Select siRNA, Ambion of Thermo Fisher Scientific) targeting human SRSF1 (siRNA ID, s12725), SRSF2 (siRNA ID, s12728), SRSF4 (siRNA ID, s12735), SRSF6 (siRNA ID, s12741), SRSF7 (siRNA ID, s12743), SRSF8 (siRNA ID, s21495), SRSF9 (siRNA ID, s16546), SRSF10 (siRNA ID, s57794), SRSF11 (siRNA ID, s17770), SRSF12 (siRNA ID, s43932), CLK1 (siRNA ID, s3162), CLK2 (siRNA ID, s3165), CLK3 (siRNA ID, s3169), CLK4 (siRNA ID, s32988), DYRK1A (siRNA ID, s4400), DYRK1B (siRNA ID, s17490), DYRK2 (siRNA ID, s16027), and DYRK3 (siRNA ID, s16026) and a non-specific control (si-NS) (siRNA ID, 4390843) as a negative control. For simultaneous knockdown of CLK isoforms and DYRK isoforms, mixture of siRNAs targeting CLK1, 2, 3, and 4 and DYRK1A, 1B, 2, and 3 were transfected, respectively. siRNA transfection was conducted using Lipofectamine RNAiMAX (Thermo Fisher Scientific) with a final concentration of each siRNAs in the culture medium set to 100 nM. Knockdown effects at the protein level were confirmed for siRNAs targeting SRSF6, CLK1, CLK2, CLK3, and CLK4 (Supplementary Fig. 6).

**Dual-fluorescence reporter system for splicing evaluation**. We previously constructed the SPREADD reporter system for *IKBKAP* IVS20 + 6 T > C[17], in which the splicing extent of *IKBKAP* exon 20 was quantified by the ratio of green fluorescent protein (GFP) and red fluorescent protein (RFP). For the reporter assay, the *IKBKAP* IVS20 + 6 T > C splicing reporter vector was transfected into HeLa cells ($0.1 \times 10^6$ cells/well, seeded in a 12-well plate) with Lipofectamine 2000 transfection reagent (Thermo Fisher Scientific) and changed to fresh media at 4–5 h after transfection to be incubated with or without compounds for 24 h before sample preparation or fixation with 4% paraformaldehyde in phosphate-buffered saline (pH 7.4). Fixed cells were then scanned with a BZ-X710 fluorescence microscope (Keyence, Osaka, Japan), and analyzed for fluorescence intensities with a hybrid-cell count software of BZ-X analyzer (1.4.0.1) (Keyence). The resulting GFP/RFP fluorescence ratio was then plotted to indicate the relative extent of *IKBKAP* exon 20 inclusion.

**RNA pull-down assay and Western blot**. Lysate for HeLa cell or primary fibroblasts lysate was prepared in lysis buffer (10 mM Tris-HCl; pH 7.4), 150 mM NaCl, 1 mM ethylenediaminetetraacetic acid, 1% Triton X-100, 0.1% sodium dodecyl sulfate, 0.25% sodium deoxycholate, and 10% glycerol with protease-inhibitor (Nacalai Tesque) and phosphatase-inhibitor (Sigma Aldrich)), followed by sonication, treatment with DNase I (Promega, Madison, WI) at 37 °C for 5 min, and centrifugation ($18,500 \times g$ at 4 °C for 15 min). The supernatant was used as a soluble fraction for RNA pull-down assay and western blot. For western blot, soluble fraction was dissolved in Laemmli buffer and separated in 5–20% Tris-glycine polyacrylamide gel (FUJIFILM Wako Pure Chemical). Separated proteins were then transferred to the PVDF membrane, and processed for blocking with 5% skim milk in Tris-buffered saline and incubation with antibody. For RNA pull-down assay, 5′-biotin, 3′-dTdT-attached RNAs (5′-UUGGACAAGUAA-GUGGCCAUUGUACU-3′ (oAM153) for the *IKBKAP* exon 20 donor region of reference sequence, 5′-UUGGACAAGUAAGCGCCAUUGUACU-3′ (oAM154) for the *IKBKAP* exon 20 donor region with IVS20 + 6 T > C mutation, 5′-UUGUACUGUUUGCGACUAGUUAGCU-3′ (oAM155) for the site-a, 5′-GUUAGCUUGUGAUUUAUGUGUGAAG-3′ (oAM156) for the site-b, 5′-AAGACAAUAAGUAUUUUAUUACAAU-3′ (oAM209) for the site-c, and 5′-GUUAGCUUGUGAUUUAUAUACAAAG-3′ (oAM167) for the site-b(mt)) were incubated with HeLa cell lysate (for oAM155, oAM156, oAM209, and oAM167) or HeLa nuclear extract (for oAM153 and oAM154) in presence of 1% DMSO or 10 μM RECTAS (CILBIOTECH, s.a., Mons, Belgium) for 16 h with rotation at 4 °C, followed by washing with Tris-buffered saline three times and eluting with Laemmli buffer. Eluted proteins were then analyzed by Western blotting. Detection was conducted with ChemiDoc MP Imaging System (Bio-Rad) and analyzed with Image J (1.52a)[65].

**Compound pull-down assay**. b-RECTAS was synthesized as a RECTAS analog with *IKBKAP* exon 20-targeting activity with covalent biotin labeling. The resulting b-RECTAS was used for the compound pull-down assay with neutravidin beads. Neutravidin beads were incubated with 0, 100, or 200 nM of biotin-conjugated RECTAS for 2 h at 4 °C. Following washing with Tris-buffered saline (pH 7.4), the biotin-RECTAS-bound beads were then incubated with HeLa cell lysate or GST-CLK1 recombinant protein expressed in Sf9 insect cells (ProQinase, Freiburg, Germany) for 16 h at 4 °C. For competition assays, 100 nmol RECTAS was

incubated during the protein binding step, followed by washing and eluting with Laemmli buffer. Eluted samples were analyzed by western blotting.

**Cell viability assay**. For cell viability assay, HeLa cells were seeded in a 96-well plate in triplicate and incubated with compounds for 72 h. Cell viability was measured by WST-8 assay using Cell Counting Kit-8 (Dojindo, Kumamoto, Japan). Absorbance at 450 nm was measured with ARVO X5 multimode plate reader (Perkin Elmer, Waltham, MA).

**RNA-Seq**. Total RNA was extracted from FD patient fibroblasts or DRG from *IKBKAP* (IVS20 + 6 T > C) transgenic mice, using TRIzol (Thermo Fisher Scientific) with DNase treatment. For FD patient fibroblasts, total RNA (2.5 μg) was poly-A purified with a Dynabeads mRNA DIRECT Micro Kit (Thermo Fisher Scientific), and used for library preparation with an Ion Total RNA-seq Kit v2 (Thermo Fisher Scientific). Single-end sequencing was performed on an Ion Proton System (Thermo Fisher Scientific). For DRG from *IKBKAP* (IVS20 + 6 T > C) transgenic mice, total RNA (0.6–0.7 μg) was poly-A purified and the library was prepared with TruSeq stranded mRNA library kit (Illumina, Inc. San Diego, CA). RNA-Seq was then conducted with NovaSeq 6000 (Illumina, Inc.), and Real-Time Analysis (RTA) 1.9 software (Illumina, Inc.) was applied for primary data processing. Sequence reads were split into 60-base fragments from the 5′ end and mapped to the human genome (GRCh38) with STAR (ver. 2.4.1d), using the Ensemble annotation GRCh38.94 as reference.[66] Sequence reads with average Phred quality scores lower than 17, and those from rRNA, tRNA, snRNA, snoRNA, or repetitive elements in the Repbase database[67] were eliminated. To calculate the PSI values, we used all internal exons from representative variants for each gene selected based on the nucleotide sequences of variants and coding sequences in UniProt[68]. The PSI value of the exon ($n$) was calculated as the number of junction reads of exon ($n − 1$) and ($n$) divided by the sum of junction reads of exon ($n − 1$) and ($n$) and exon ($n − 1$) and ($n + 1$). For analysis of FD patient fibroblasts, PSI values obtained from patient fibroblasts P1 and P2 were averaged, and exons with more than 15 reads per million junction read for the sum of their skipping forms and inclusion forms in DMSO and/or RECTAS-treatment conditions were used for the analysis. The Sashimi plots were generated using an Integrated Genomic Viewer (ver. 2.8.0)[69]. The RNA-seq data were deposited at the National Bioscience Database Center database for patient fibroblasts (accession: hum0180), and at the Gene Expression Omnibus of the National Center for Biotechnology Information for transgenic mice (accession: GSE161109).

**Statistics**. The student's *t* test was used to determine the statistical difference between the two groups, and P values <0.05 were considered statistically significant. Regression analysis for determination of $EC_{50}$ and $CT_{50}$ was conducted by GraphPad Prism 7.05. (GraphPad Software, San Diego, CA).

**Reporting summary**. Further information on research design is available in the Nature Research Reporting Summary linked to this article.

## Data availability
All relevant data are available from the authors. The original RNA-seq data were deposited at the National Bioscience Database Center database with the accession ID hum0180 for patient fibroblasts, and at the Gene Expression Omnibus of the National Center for Biotechnology Information for transgenic mice with the accession ID GSE161109. The Repbase database is available at https://www.girinst.org/repbase/. GRCh38 human reference genome and Ensemble annotation GRCh38.94 are available from the Ensembl database (https://ensembl.org). Source data are provided with this paper.

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

## Acknowledgements

We acknowledge Dr. Shuh Narumiya of Kyoto University Graduate School of Medicine for his encouragement and suggestions throughout the work, members of the Department of Anatomy and Developmental Biology and the Department of Drug Discovery Medicine of Kyoto University Graduate School of Medicine for fruitful discussions, and Ms. Chihiro Takashima and Ms. Chizuru Tsuzuki for technical supports. This study was supported by 15H05721 (to M.A., T.A., and M.H.) and 19K07367 (to M.A.) from the Japan Society for the Promotion of Science; JP18kk0305003h0003, 19ek0109327h0002, and 19lm0203054h0002 of the Japan Agency for Medical Research and Development (to M.A., T.A., and M.H.). A.R.K. and Y.J.K. acknowledge support from NIH grants R37-GM42699, F30HL137326-03, and the St. Giles Foundation.

## Author contributions

M.A., T.A., Y.J.K., R.K., S.M., S.S., and T.S conducted biological experiments. K.I. and M.D. conducted and analyzed RNA-seq. N.T. conducted compound synthesis. R.S. provided patient iPS cells. A.R.K. provided technical interpretation. M.A. and M.H. wrote the paper.

## Competing interests

S.M. and S.S. are employees of Kyorin Pharmaceutical Co., Ltd. M.H. is a founder, shareholder, and member of the scientific advisory board of KinoPharma, Inc., and BTB Drug Development Research Center Co., Ltd. The remaining authors declare no competing interests.
