## [Peer Review File · Nature Communications]

Reviewers' Comments:

Reviewer #1:

Remarks to the Author:

In this manuscript, Masahiko Ajiro et al. describe the mechanism of RECTAS modulating FD-associated IKBKAP exon 20 splicing. The group previously identified RECTAS from a chemical library correcting abnormal splicing of IKBKAP mutation and suggested RBM24 as the RECTAS effector. In their new study, they found no change in RBM24 level or activity after RECTAS treatment. They identified SRSF6 KD reduced RECTAS's effect on IKBKAP reporter and confirmed SRSF6 binding of specific cis element in intron 20. The authors further showed RECTAS interacts with CLK to affect splicing. Finally, they tested RECTAS in human patient ipSC derived sensory neurons and humanized fd mouse model and showed its effect in enhancing exon 20 splicing. Overall, this is an exciting story proposing a new mechanism involving CLK and SRSF6 to explain RECTAS's activity on exon 20 splicing. CLK and SRSF6 are widely expressed and presumably control many targets. RECTAS' limited impact of on transcriptome is therefore surprising but this can be a future study. How RECTAS activates CLK is also important but probably needs structural analysis and would be for future studies. The ipsc and in vivo mouse tests are significant results. The manuscript will be significantly improved if the authors validate their proposed mechanism in these systems.

Major comment:

1. As the authors showed RECTAS can act through RBM24 and CLK-SRSF6 (in different cell types), it is important to know whether and which pathway is relevant in disease-relevant ipsc and mouse models. For example, does RECTAS affect SRSF6 phosphorylation and or expression in drg neurons and ipsc-SNs? Do SRSF6 and CLK KD reduce RECTAS's activity in these cells?
2. I probably miss this. The authors only showed the effect of SRSF6 and CLK on the exon 20 reporter. Do they (knockdown) affect splicing of exon 20 expressed from the endogenous locus? Ideally, this should be tested in wt vs ikbkap mutant cells.
3. The connection between CLK and SRSF6 under RECTAS is suggested but without proof. Can RECTAS still enhance SRSF6 phosphorylation in the absence of CLK? Can SRSF6 overexpression abrogate CLK1 KD in controlling exon 20 splicing?
4. Fig3e is probably the most important data so far supporting the mechanistic link between clk and srsf6 but is not explained in the text or figure legend. How long is the treatment? What is the kinetics of srsf6 phosphorylation? how long can the enhanced phosphorylation sustain after treatment?
5. Similarly in fig4b: how long does the RECTAS treatment take to increase splicing in patient cells? How long can the splicing correction stay after treatment?
6. Western blots are needed to confirm SRSF6 and CLK KD efficiency.
7. Statistics is missing for figures 2e, 2f, 4b. Throughout all figures, please clearly indicate numbers of biological replicates

Minor comment:

1. Fig4c, can the authors show the splicing gel of the endogenous exon 20? Please clarify whether fig4d is based on fluorescence or RNA.
2. Figure legend needs to indicate cell types used for fig1d-e.

Reviewer #2:

Remarks to the Author:

In this article, authors reported that RECTAS enhances inclusion of IKBKAP-FD exon 20 through the phosphorylation of SRSF6 and that oral administration of RECTAS to a transgenic model mouse of FD enhanced exon 20 inclusion in dorsal root ganglia. They proposed RECTAS treatment for FD. This article was well written. Before acceptance for the publication, it is necessary to clarify the following points.

1. In the introduction section, authors described that "RECTAS directly affects mRNA splicing, and

our transcriptome analyses revealed that RECTAS affects the splicing of only a limited set of genes, suggesting that it has high specificity and a direct role in correcting IKBKAP aberrant splicing.⁸ This sentence means that important information of RECTAS is described in the reference 8 reported by Ohe et al. However, the word of "RECTAS" cannot be identified in the reference 8 text. This raises a big question on this study.

2. Authors have already reported that RBM24 enhances inclusion of exon 20. Here, they showed that RECTAS also enhances inclusion of exon 20. The relationship between RBM24 and SRSF6 is not unclear. RBM24 binds to a nucleotide region from +13 to +29, whereas SRSF6 binds to a region from +45 to +50. Since RBM24 binding site is located nearer to the splice site, the binding of RBM24 should be taken into the consideration for splicing regulation.

3. Rubin and Anderson listed 10 compounds that have ability to enhance exon 20 inclusion (Appl Clin Genet, 2017). Superiority of RECTAS as a treatment compound for FD treatment should be discussed.

4. In treatment of the transgenic mouse with RECTAS, authors revealed exon 20 inclusion in dorsal root ganglia. It is important to know off-target effect of RECTAS that activates SRSF6 phosphorylation. Are there any splicing abnormalities in the treated mouse?

Reviewer #3:

Remarks to the Author:

In this manuscript, Masahiko Ajiro and collaborators investigated the mode of action of the molecule RECTAS on the splicing of IKBKAP-FD exon 20, which contains a suboptimal 5' splice site. Indeed, a T to C mutation in the 5' splice site creates a mismatch in the RNA duplex formed between this sequence and the U1 snRNA upon recruitment of the spliceosome. As a result, the exon 20 is skipped, which generates a premature termination codon leading to the reduced expression of the encoded protein. The T to C mutation is responsible for more than 99.5% of cases of familial dysautonomia (FD). The authors found that the molecule RECTAS interacts with cdc-like kinases and enhances the phosphorylation of the SR protein SRSF6. This phosphorylation enhancement of SRSF6 forces the protein to promote the splicing of the IKBKAP-FD exon 20. The ISE bound by SRSF6 was identified and the molecule could restore exon 20 inclusion in patient induced pluripotent stem cells-derived sensory neurons and transgenic mice. The data are very impressive and the fact that a molecule could activate kinases is very interesting as it increases the panel of strategies that could be used against diseases coming from splicing defects.

Major points

- Nevertheless, the effect of the molecule on SRSF6 phosphorylation is very weak (Fig. 3f) and the fact that a molecule stimulates the activity of an enzyme is not common. I understand that investigating the mode of action of the molecule may be outside of the scope of this paper. However, these data would be more convincing if they could be confirmed by some in vitro experiments. An experiment following the phosphorylation of SRSF6 by CLK in the presence or not of the molecule and a direct evidence that this phosphorylation enhances the affinity of the protein for the ISE would be required.

- In the introduction, the authors mentioned small compounds promoting SMN2 exon7 inclusion but do not refer to the paper that really explains the mode of action of the molecule: Campagne et al., 2019 (PMID: 31636429). Can the authors completely rule out the hypothesis of a possible direct contribution of the molecule in the stabilization of the U1-5'SS duplex?

Response to referees

Reviewer #1 (Remarks to the Author):

In this manuscript, Masahiko Ajiro et al. describe the mechanism of RECTAS modulating FD-associated IKBKAP exon 20 splicing. The group previously identified RECTAS from a chemical library correcting abnormal splicing of IKBKAP mutation and suggested RBM24 as the RECTAS effector. In their new study, they found no change in RBM24 level or activity after RECTAS treatment. They identified SRSF6 KD reduced RECTAS's effect on IKBKAP reporter and confirmed SRSF6 binding of specific cis element in intron 20. The authors further showed RECTAS interacts with CLK to affect splicing. Finally, they tested RECTAS in human patient ipSC derived sensory neurons and humanized fd mouse model and showed its effect in enhancing exon 20 splicing. Overall, this is an exciting story proposing a new mechanism involving CLK and SRSF6 to explain RECTAS's activity on exon 20 splicing. CLK and SRSF6 are widely expressed and presumably control many targets. RECTAS' limited impact of on transcriptome is therefore surprising but this can be a future study. How RECTAS activates CLK is also important but probably needs structural analysis and would be for future studies. The ipsc and in vivo mouse tests are significant results. The manuscript will be significantly improved if the authors validate their proposed mechanism in these systems.

(Response)

We appreciate the reviewer for providing thoughtful comments and suggestions. We believe our current study will provide a significant advance in therapeutic strategy for familial dysautonomia, based on results in sensory neurons from patient and transgenic mice, along with target exon-selectivity and pre-clinical profile of RECTAS. We made additional validations according to reviewer's suggestions, and we believe that the manuscript became more convincing through this revision.

Major comment:

1. As the authors showed RECTAS can act through RBM24 and CLK-SRSF6 (in different cell types), it is important to know whether and which pathway is relevant in disease-relevant ipsc and mouse models. For example, does RECTAS affect SRSF6 phosphorylation and or expression in drg neurons and ipsc-SNs? Do SRSF6 and CLK KD reduce RECTAS's activity in these cells?

(Response)

We appreciate the reviewer for raising important inquiries. In our previous study (Ohe K. et al. *RNA* (2017) 23:1393-1403), we revealed *RBM24* is a tissue-dependent splice enhancer for *IKBKAP* exon 20 inclusion in skeletal muscle cells, and expression of *RBM24* is exclusively seen in skeletal muscle and heart tissues. Indeed, we checked *RBM24* expression in RECTAS-responding cells and tissues (HeLa cells, patient primary fibroblasts, and mouse DRG), and found that there is no detectable *RBM24* expression in these cells or tissues (Fig. 1d and Fig. 4h in the revised manuscript). Moreover, RECTAS treatment also did not affect *RBM24* expression in these cells or tissues. From these observations, we ruled out relevance of *RBM24* in RECTAS-induced *IKBKAP* exon 20 inclusion, and concluded CLK-SRSF6 serves as a major pathway in these RECTAS-responding cells. Consistently, we also confirmed treatment of iPSC-SNs by pan-CLK inhibitor CaNDY (Shibata S. et al. *Cell Chem. Biol.* 2020) reduced RECTAS's activity (Fig. 4c of the revised manuscript). The manuscript was also revised accordingly (**line 77-80, 95-100, 208-210, 229-232**).

Figure. 1

(d) RT-PCR for *RBM24* expression in human tissues (liver, kidney, lung, heart, and skeletal muscle) and cell lines (HeLa cells and FD patient primary fibroblasts (P1), treated with 10 μ M RECTAS or 0.1% DMSO for 24 h). *ACTB* served as a loading control.

Figure. 4

(h) RT-PCR for *Rbm24* in blood cells, skeletal muscle, and liver tissues from 2 months-old B6 mouse, as well as DRG from MOCK or RECTAS (300 mg/kg BW)-administered *IKBKAP*-FD transgenic mice. *Actb* served as a loading control.

Figure. 4

(c) RT-PCR for *IKBKAP* exon 19-21 and *ACTB* as a loading control for iPSC-SNs (day 12) treated with CaNDY (10 μ M) or 0.1% DMSO for 2h, followed by addition of RECTAS (10 μ M) or 0.1% DMSO in media and incubation for 22 h.

2. I probably miss this. The authors only showed the effect of SRSF6 and CLK on the

exon 20 reporter. Do they (knockdown) affect splicing of exon 20 expressed from the endogenous locus? Ideally, this should be tested in wt vs ikbkap mutant cells.

(Response)

We appreciate the reviewer for pointing out the endogenous validation, as well as inclusion of control fibroblasts without *IKBKAP* mutation. In the revised manuscript, we confirmed effect of SRSF6 and CLK knockdown in primary fibroblasts from FD patients (Fig. 2h and S3a in the revised manuscript). We observed consistent exon 20 skipping in SRSF6 or CLK knocked-down FD patient primary fibroblasts. We also obtained two lines of primary fibroblasts isolated from healthy individuals without *IKBKAP* IVS20 T>C mutation, and verified knockdown and CLK inhibitor effects. We validated there is no exon 20 skipping detectable by RT-PCR from control fibroblasts both after SRSF6 or CLK knockdown (Fig. 2j and S3b) and CLK inhibitor treatments (Fig. 2i and S2b). These data further confirmed that a suboptimal donor site due to the IVS20+6T>C mutation creates dependency on CLK and SRSF6. The manuscript was also revised accordingly (line 159-168).

Figure. 2

(h) RT-PCR for *IKBKAP*-FD exon 19-21 for primary fibroblasts from FD patient (P1), transfected with non-targeting siRNA (si-NS), siRNAs for SRSF6 (si-SRSF6), or CLK isoforms (si-CLKs) for 72 h. (j) RT-PCR for *IKBKAP*-FD exon 19-21 for primary fibroblasts from healthy individual (C1), transfected with si-NS, si-SRSF6, or si-CLKs for 72 h.

Supplementary Figure S2. Knockdown effect of SRSF6 and CLK isoforms on *IKBKAP* exon 20 splicing in primary fibroblasts from FD patient and healthy donor. The knockdown effect of SRSF6 (si-SRSF6) and CLK isoforms (si-CLKs) for endogenous *IKBKAP*-FD exon 20 was shown for primary fibroblasts from FD patient (P2) (a) and healthy donor (C2) (b). Non-targeting siRNA (si-NS) served as a negative control, and *ACTB* served as a loading control.

Figure. 2

(i) RT-PCR for *IKBKAP*-FD exon 19-21 for primary fibroblasts from healthy individual (C1) treated with the indicated compounds (2 or 10 μM) or solvent only (0.1 % DMSO) for 24 h.

Supplementary Figure S1. Treatment with RECTAS, CLK inhibitors, and SRPK inhibitors in primary fibroblasts from healthy donor.

Compound response (24 h treatment at 2 or 10 μM) of endogenous *IKBKAP*-FD exon 20 was shown for primary fibroblasts from healthy donor (C2). Solvent only (0.1% DMSO) served as a negative control, and *ACTB* served as a loading control.

3. The connection between CLK and SRSF6 under RECTAS is suggested but without proof. Can RECTAS still enhance SRSF6 phosphorylation in the absence of CLK? Can SRSF6 overexpression abrogate CLK1 KD in controlling exon 20 splicing?

(Response)

We thank the reviewer for raising this question. We explored relevance of CLK and SRF6 under RECTAS by analyzing *IKBKAP* exon 20 inclusion in cells with CLK inhibition. We recently reported CaNDY as a pan-CLK inhibitor (Shibata S. et al. *Cell Chem. Biol.* 2020), effective to all CLK isoforms (CLK1-4). In the absence of intracellular CLK activity by CaNDY treatment, we found RECTAS cannot induce exon 20 inclusion in HeLa cells (Fig. 2e in the revised manuscript), and the same was true for SRSF6 overexpression (Fig. 2f in the revised manuscript). Explanations for these observations were added in the revised manuscript (**line 151-159**).

Figure. 2

(e-f) *IKBKAP*-FD exon 20 inclusion rates were quantified by the GFP/RFP ratio in *IKBKAP*-FD reporter in HeLa cells with or without CLK inhibition by CaNDY treatment (10 μM , 24 h), in response to RECTAS treatment (2 μM , 10 μM , or 0.1% DMSO only for 24 h) (e) or transfection with SRSF6 expression vector (SRSF6) or empty vector (control) for 24 h (f).

4. Fig3e is probably the most important data so far supporting the mechanistic link between *clk* and *srsf6* but is not explained in the text or figure legend. How long is the treatment? What is the kinetics of *srsf6* phosphorylation? how long can the enhanced phosphorylation sustain after treatment?

(Response)

In the revised manuscript, we included time-course data for SRSF6 phosphorylation upon RECTAS treatment (Fig. 3g in the revised manuscript). This effect is transient, with evident at 6h after the treatment and return to basal level 10 h after the washout. In addition, time point (24 h) for Fig. 3f was also indicated in the revised manuscript (line 182-185).

Figure. 3

(g) Phosphor SRSF6 in FD fibroblasts (P1) was quantified at 0, 6, 10, 24 h after RECTAS (10 μ M) treatment (columns, 0h, 6h, 10h, and 24h), as well as 10 h after washout following RECTAS treatment (10 μ M) for 24 h (column, wo). Lamin B1 served as a loading control.

5. Similarly in fig4b: how long does the RECTAS treatment take to increase splicing in patient cells? How long can the splicing correction stay after treatment?

(Response)

As pointed out by the reviewer, we performed time-course analysis for *IKBKAP* exon 20 splicing following RECTAS treatment in iPSC-SNs in the revised Supplementary Figure S3. We observed inclusion of *IKBKAP* exon 20 in FD patient iPSC-SNs beginning around 2 h after the incubation and lasting for ~4 h after washout. These data indicate RECTAS corrects *IKBKAP*-FD exon 20 inclusion in a transient manner. Manuscript was also revised accordingly (line 206-208).

Supplementary Figure S3. Time-course experiment for exon 20 inclusion promoting activity by RECTAS.

FD patient cell-derived iPSC-SNs at 12 days of induced differentiation were analyzed for *IKBKAP* exon 20 splicing at 0, 2, 4, 6, and 24 h after the RECTAS treatment (10 μ M). 24 h-RECTAS treated iPSC-SNs were also subjected to washout by PBS (-) and kept in culture media without RECTAS, and analyzed at 0, 2, 4, 6, and 24 h after the washout. *ACTB* served as a loading control. Representative data from three iPSC strains are shown. PSI, percent spliced-in; E20 (+) and E20 (-), exon 20 inclusion and skipping products, respectively.

6. Western blots are needed to confirm SRSF6 and CLK KD efficiency.

(Response)

We appreciate the reviewer for providing the comment. We accordingly examined Western blots for SRSF6 and CLK1-4 to confirm knockdown efficiency by siRNAs we used (Supplementary Fig. S6 of the revised manuscript). The manuscript was also revised accordingly (**line 414-415**).

Supplementary Figure S6. Western blot analysis for knockdown efficiency.

HeLa cells were transfected with non-targeting siRNA (si-NS) or siRNA targeting SRSF6, CLK1, CLK2, CLK3, and CLK4. Forty-eight hours after the transfection, each protein expression was analyzed by Western blot. Fold change was based on densitometry analysis by Image J 1.x.

7. Statistics is missing for figures 2e, 2f, 4b. Throughout all figures, please clearly indicate numbers of biological replicates

(Response)

We appreciate the comments. In every figure in the revised manuscript, we clarified replicate information in corresponding figure legends following the guideline.

Minor comment:

1. Fig4c, can the authors show the splicing gel of the endogenous exon 20? Please clarify whether fig4d is based on fluorescence or RNA.

(Response)

We thank the reviewer for raising this question. In previous version of the manuscript, *IKBKAP* exon 20 splicing in neuro 2A cells were examined only by fluorescence from reporter vectors. We thus repeated these experiments to validate at RNA levels for both exogenous and endogenous transcripts. As shown in the revised Fig. 4f, we confirmed RECTAS-induced exon 20 inclusion for *IKBKAP* (IVS20+6T>C), but no effect in homologous mouse endogenous *Ikbkap* without mutation (Fig. 4f and Supplementary Figure S4). Explanations in the manuscript was revised accordingly (line 215-218).

Figure. 4

(f) RT-PCR for exogenous human *IKBKAP* (oAM124+oAM126, primers designed for vector backbone) and endogenous mouse *Ikbkap* (oAM666+oAM667, primers designed for exon 18 and 20) in neuro 2A cells transfected with *IKBKAP*-WT or -FD reporter and treated with 2 or 10 μ M RECTAS or 0.1% DMSO for 24 h. *Actb* served as a loading control.

Supplementary Figure S4. Diagram for human *IKBKAP* and mouse *Ikbkap*.

Exon 18, 19, and 20 of mouse *Ikbkap* is highly homologous to exon 19, 20, and 21 of human *IKBKAP* with exact match in exon length and surrounding intronic sequences.

2. Figure legend needs to indicate cell types used for fig1d-e.

(Response)

We appreciate the reviewer for pointing out. Reporter assays, reassigned as Fig. 1f-g in the revised manuscript, were conducted in HeLa cells. The legends were revised to indicate cell types used (**line 538-540**).

Reviewer #2 (Remarks to the Author):

In this article, authors reported that RECTAS enhances inclusion of IKBKAP-FD exon 20 through the phosphorylation of SRSF6 and that oral administration of RECTAS to a transgenic model mouse of FD enhanced exon 20 inclusion in dorsal root ganglia. They proposed RECTAS treatment for FD.

This article was well written. Before acceptance for the publication, it is necessary to clarify the following points.

(Response)

We appreciate comments and suggestions provided by the reviewer. We conducted additional studies according to reviewer's suggestions. We believe evidences obtained through this revision reinforced therapeutic potential and mechanistic insight of RECTAS for *IKBKAP* mis-splicing.

1. In the introduction section, authors described that "RECTAS directly affects mRNA splicing, and our transcriptome analyses revealed that RECTAS affects the splicing of only a limited set of genes, suggesting that it has high specificity and a direct role in correcting IKBKAP aberrant splicing.⁸" This sentence means that important information of RECTAS is described in the reference 8 reported by Ohe et al. However, the word of "RECTAS" cannot be identified in the reference 8 text. This raises a big question on this study.

(Response)

We appreciate the reviewer for pointing out this. We noticed reference number at this citation was shifted accidentally in the former version. Yoshida M. et al. *PNAS* (2015) 112(9): 2764-2769 is the correct citation, in which we reported exon array analysis for RECTAS treatment. All references were carefully checked through this revision.

2. Authors have already reported that RBM24 enhances inclusion of exon 20. Here, they showed that RECTAS also enhances inclusion of exon 20. The relationship between RBM24 and SRSF6 is not unclear. RBM24 binds to a nucleotide region from +13 to +29, whereas SRSF6 binds to a region from +45 to +50. Since RBM24 binding site is located nearer to the splice site, the binding of RBM24 should be taken into the consideration for splicing regulation.

(Response)

In the revised manuscript, we added RBM24 expression data for RECTAS-responding cells and tissues (HeLa cells, primary fibroblasts, and mouse DRG) (Fig. 1d and 4h). As we and others reported previously, RBM24 expression is restricted mainly in skeletal muscle and heart tissues (Ohe K. et al. *RNA* (2017) 23:1393-1403). In fact, our data in Fig. 1d and Fig. 4h of the revised manuscript indicates RECTAS-responding cells and tissues are negative for RBM24 expression, and also there was no induction of RBM24 upon RECTAS treatment or administration. These data rule out involvement of RBM24 in enhanced exon 20 inclusion by RECTAS.

On the other hand, considering the fact that binding site of RBM24 (+13 to +29) is close to that of SRSF6 (+45 to +50), we also speculate that effect of RECTAS could be interfered by RBM24 in skeletal muscle and cardiomyocytes, although these tissues show higher basal inclusion rate of exon 20 than RBM24-negative neural tissues (Ohe K. et al. *RNA* (2017) 23:1393-1403).

The manuscript was revised to include above information (**lines 77-80, 95-100, 127-129, 229-232**).

Figure. 1

(d) RT-PCR for *RBM24* expression in human tissues (liver, kidney, lung, heart, and skeletal muscle) and cell lines (HeLa cells and FD patient primary fibroblasts (P1), treated with 10 μ M RECTAS or 0.1% DMSO for 24 h). *ACTB* served as a loading control.

Figure. 4

(h) RT-PCR for *Rbm24* in blood cells, skeletal muscle, and liver tissues from 2 months-old B6 mouse, as well as DRG from MOCK or RECTAS (300 mg/kg BW)-administered *IKBKAP*-FD transgenic mice. *Actb* served as a loading control.

3. Rubin and Anderson listed 10 compounds that have ability to enhance exon 20 inclusion (Appl Clin Genet,2017). Superiority of RECTAS as a treatment compound for FD treatment should be discussed.

(Response)

We appreciate the reviewer for providing suggestions. According to Rubin Y. and Anderson S.L., Appl. Clin. Genet. (2017) 10: 95-103, we obtained following compounds, kinetin, digoxin, daidzein, δ -tocotreinol, genistein, (-)-epigallocatechin gallate (EGCG), phosphatidyl serine, bortezomib, cafilzomib, and epoxomicin, and tested for *IKBKAP*-FD exon 20 inclusion activity and cytotoxicity in our splicing reporter system in HeLa cells. As a result, we confirmed that RECTAS indicates the lowest 25% effective concentration (EC_{25}) (0.72 μ M) for exon 20 inclusion activity among compounds we tested. Two other compounds, kinetin and EGCG, reached EC_{25} of RECTAS but 15.5- and 241-fold higher than that of RECTAS. 50% cytotoxicity (CT_{50}) also confirmed less toxic property of RECTAS with >20 μ M of CT_{50} . Above results were included as Fig. 2k,l,m,n in the revised manuscript, and manuscript was revised accordingly (**line 168-172, 272-275, 362-367**).

k

Compound	EC ₂₅ (μM)	CT ₅₀ (μM)
RECTAS	0.72	> 20
kinetin	11.19	> 20
digoxin	> 20	0.07
daidzein	> 300	10
δ-tocotreinol	> 300	62
genistein	> 300	8
EGCG	173.36	175
PS	> 300	82
bortezomib	> 10	0.03
cafilzomib	> 10	0.03
epoxomicin	> 10	0.006

**Figure. 2**

(k) 25% effective concentration (EC₂₅) and 50% cytotoxic concentration (CT₅₀) were shown for indicated compounds. EC₂₅ was determined by *IKBKAP* (FD) reporter assay in HeLa cells with treatment for 24 h. %GFP ((GFP fluorescence intensity)/(GFP + RFP fluorescence intensity) of 20 μM RECTAS condition was set to 100%. CT₅₀ was determined in HeLa cells for 72 h treatment. (l-n) %GFP ((GFP fluorescence intensity)/(GFP + RFP fluorescence intensity) and %cell viability compared to control cells treated only with solvent are plotted for RECTAS (l), kinetin (m), and (-)-epigallocatechin gallate (EGCG) (n).

4. In treatment of the transgenic mouse with RECTAS, authors revealed exon 20 inclusion in dorsal root ganglia. It is important to know off-target effect of RECTAS that activates SRSF6 phosphorylation. Are there any splicing abnormalities in the treated mouse?

(Response)

We appreciate the reviewer for raising the question. In order to answer this point, we conducted RNA-Seq for RNAs extracted from DRGs of *IKBKAP*-FD transgenic mice with or without RECTAS administration (300 mg/kg BW). As shown in Supplementary Fig. S5a,b, and Table S2 of the revised manuscript, we found *IKBKAP*-FD showed the

13th biggest Δ PSI, consistent with a selective effect of RECTAS observed in primary fibroblasts from FD patient (Fig. 1b,c and Supplementary Table S1). We detected total 142 of altered splice events with $|\Delta$ PSI \geq 0.05, and majority of them showed smaller changes than *IKBKAP*-FD exon 20. The RNA-Seq data were deposited in Gene Expression Omnibus of National Center for Biotechnology Information with accession ID GSE161109. Moreover, most of those alternative splicing are physiologically occurring and thus effects of off-target may be limited. Indeed, pre-clinical toxicology tests in rodents indicate excess doses of RECTAS (up to 1,000 mg/kg BW) is tolerated without showing body weight changes or physiological abnormalities (Supplementary Table S3). We revised the manuscript accordingly to include above observations (**line 225-229, 475-478, 491-492, 506-508**).

Supplementary Fig. S5. RNA-Seq analysis for DRG from *IKBKAP*-FD transgenic mice treated with MOCK or RECTAS (300 mg/kg BW).

(a) Differential splice events with >0.05 of $|\Delta$ PSI ($PSI_{RECTAS} - PSI_{CMC}$) in RECTAS (300 mg/kg BW)-administered mice over CMC-administered mice were plotted. Vertical, Δ PSI; Horizontal, gene count. Arrow (orange) indicates *IKBKAP*-FD exon 20 inclusion. (b) Representative sashimi plot with junction read number was shown for MOCK and RECTAS groups. *IKBKAP* gene structure is shown on the bottom.

Reviewer #3 (Remarks to the Author):

In this manuscript, Masahiko Ajiro and collaborators investigated the mode of action of the molecule RECTAS on the splicing of *IKBKAP*-FD exon 20, which contains a suboptimal 5' splice site. Indeed, a T to C mutation in the 5' splice site creates a mismatch in the RNA duplex formed between this sequence and the U1 snRNA upon

recruitment of the spliceosome. As a result, the exon 20 is skipped, which generates a premature termination codon leading to the reduced expression of the encoded protein. The T to C mutation is responsible for more than 99.5% of cases of familial dysautonomia (FD). The authors found that the molecule RECTAS interacts with cdc-like kinases and enhances the phosphorylation of the SR protein SRSF6. This phosphorylation enhancement of SRSF6 forces the protein to promote the splicing of the IKBKAP-FD exon20. The ISE bound by SRSF6 was identified and the molecule could restore exon 20 inclusion in patient induced pluripotent stem cells-derived sensory neurons and transgenic mice. The data are very impressive and the fact that a molecule could activate kinases is very interesting as it increases the panel of strategies that could be used against diseases coming from splicing defects.

(Response)

We appreciate comments by the reviewer. We consider our current study contributes to a novel insight for the emerging field of splice-targeting small molecule compounds. Given target selective transcriptome profile, pre-clinical property, and therapeutic effects in disease models, we consider current report will stand as one of leading studies in this field. We believe development of those splice modulators provides therapeutic modality alternative to anti-sense nucleotide therapeutics, and contribute to expand future therapeutic possibility for splicing diseases.

Major points

- Nevertheless, the effect of the molecule on SRSF6 phosphorylation is very weak (Fig. 3f) and the fact that a molecule stimulates the activity of an enzyme is not common. I understand that investigating the mode of action of the molecule may be outside of the scope of this paper. However, these data would be more convincing if they could be confirmed by some in vitro experiments. An experiment following the phosphorylation of SRSF6 by CLK in the presence or not of the molecule and a direct evidence that this phosphorylation enhances the affinity of the protein for the ISE would be required.

(Response)

We appreciate the reviewer for constructive suggestions. Confirmation by in vitro system is desirable for this study. However, by far we haven't succeeded in preparing functional recombinant SRSF6 because of the insolubility. To improve the solubility, we tried attachment of tag proteins (e.g. GST) at either amino- or carboxyl-terminal. However, it interferes with function of amino-terminal RNA binding motif or carboxyl-terminal RS domain, required for RNA binding ability of SRSF6. As we understand the importance of characterization of SRSF6 function in context of suboptimal donor RNA

recognition in future study, especially when we assess consequence of RECTAS interaction to CLK1, we mentioned that in the discussion part of the revised manuscript **(line 256-258)**.

- In the introduction, the authors mentioned small compounds promoting SMN2 exon7 inclusion but do not refer to the paper that really explains the mode of action of the molecule: Campagne et al., 2019 (PMID: 31636429). Can the authors completely rule out the hypothesis of a possible direct contribution of the molecule in the stabilization of the U1-5'SS duplex?

(Response)

As the reviewer pointed out, small molecule compounds that repair 5' splice site bulge were intensively investigated for spinal muscular atrophy. Following the suggestion by the reviewer, we examined if RECTAS has similar role for *IKBKAP*-FD exon 20 donor recognition. We conducted RNA pull-down assay by biotinylated RNA of *IKBKAP* exon donor with or without IVS20+6T>C mutation for HeLa cell nuclear extract in the presence or absence of 10 μ M RECTAS, and association of U1snRNP was evaluated by Western blot for U1-70k and SmB/B' subunit of U1snRNP. The result indicated there was no significant change detectable between pull-down products with and without 10 μ M RECTAS, whereas weaker donor recognition due to IVS20+6T>C mutation was reproduced. This data indicates mode of action of RECTAS will be different from 5' splice site bulge repairing for U1snRNP association. In addition, different from *SMN2* exon 7 donor, there is no bulge site expected from base-pairing for *IKBKAP*-FD exon 20 donor (Fig. 1a). The RNA pull-down data was included as the Fig. 3h and new reference (Campagne S. et al. *Nat. Chem. Biol.* (2019) 15:1191-1198) was added in the revised manuscript. Explanations and discussion were revised accordingly **(line 48, 185-191)**.

WT exon 20 donor RNA (oAM153): 5'-UUGGACAAGUAAGUGCCAUUGUACU-3'

FD exon 20 donor RNA (oAM154): 5'-UUGGACAAGUAAGCGCCAUUGUACU-3'

(IVS20+6T>C mutation was underscored)

Figure. 3

(h) Western blot for U1-70k and SmB/B' for pull-down products by biotin-conjugated *IKBKAP* exon 20 donor regions with (oAM153) or without (oAM154) IVS20+6T>C (labeled as FD and WT, respectively). Pull-down was conducted in the presence of 10 μ M RECTAS with 1% DMSO or 1% DMSO only. RNA (-), control sample without RNA bait. Input, HeLa nuclear extract of 5% input amount.

Yours sincerely,

Masatoshi Hagiwara, MD, PhD

Professor/Vice-provost, Kyoto University Graduate School of Medicine

Department of Anatomy and Developmental Biology,

Kyoto University Graduate School of Medicine

Building C, 3rd Floor, Yoshida-Konoe-cho, Sakyo-ku, Kyoto 606-8501, Japan

hagiwara.masatoshi.8c@kyoto-u.ac.jp

Tel: +81-75-753-4341 or 4333, Fax: +81-75-751-7529

Lab web site: <http://www.anat1dadb.med.kyoto-u.ac.jp/>

Reviewers' Comments:

Reviewer #1:

Remarks to the Author:
no additional comments

Reviewer #2:

Remarks to the Author:

Authors well responded to the reviewer's comments. This revised manuscript is acceptable.

Reviewer #3:

Remarks to the Author:

we are happy with this revised version that did address our criticisms.